# Phase-shifting the circadian glucocorticoid profile induces disordered feeding behaviour by dysregulating hypothalamic neuropeptide gene expression

Mitsuhiro Yoshimura[1,2], Benjamin P. Flynn [1], Yvonne M. Kershaw[1], Zidong Zhao[1], Yoichi Ueta [2], Stafford L. Lightman [1,3] & Becky L. Conway-Campbell [1,3✉]

Here we demonstrate, in rodents, how the timing of feeding behaviour becomes disordered when circulating glucocorticoid rhythms are dissociated from lighting cues; a phenomenon most commonly associated with shift-work and transmeridian travel 'jetlag'. Adrenalectomized rats are infused with physiological patterns of corticosterone modelled on the endogenous adrenal secretory profile, either *in-phase* or *out-of-phase* with lighting cues. For the *in-phase* group, food intake is significantly greater during the rats' active period compared to their inactive period; a feeding pattern similar to adrenal-intact control rats. In contrast, the feeding pattern of the *out-of-phase* group is significantly dysregulated. Consistent with a direct hypothalamic modulation of feeding behaviour, this altered timing is accompanied by dysregulated timing of anorexigenic and orexigenic neuropeptide gene expression. For *Neuropeptide Y* (*Npy*), we report a glucocorticoid-dependent direct transcriptional regulation mechanism mediated by the glucocorticoid receptor (GR). Taken together, our data highlight the adverse behavioural outcomes that can arise when two circadian systems have anti-phasic cues, in this case impacting on the glucocorticoid-regulation of a process as fundamental to health as feeding behaviour. Our findings further highlight the need for development of rational approaches in the prevention of metabolic dysfunction in circadian-disrupting activities such as transmeridian travel and shift-work.

[1] Translational Health Sciences, Bristol Medical School, University of Bristol Dorothy Hodgkin Building, Whitson Street, Bristol BS1 3NY, UK. [2] Department of Physiology, University of Occupational and Environmental Health, Japan 1-1 Iseigaoka, Yahatanishi-ku, Kitakyushu 807-8555, Japan. [3] These authors jointly supervised this work: Stafford L. Lightman and Becky L. Conway-Campbell. ✉email: b.conway-campbell@bristol.ac.uk

Secretion of adrenal glucocorticoids (corticosterone (CORT) in rats and cortisol in humans) is regulated by the hypothalamic-pituitary-adrenal (HPA) axis, and characterised by both circadian and ultradian rhythmicity. These daily and hourly fluctuations in adrenal glucocorticoid secretion result in rapid changes of circulating hormone levels in the blood and within target tissues including the brain[1,2]. Ultradian glucocorticoid pulse amplitude rises steeply in the hours before waking, in an 'anticipatory' response to the active phase, then gradually diminishes throughout the day reaching a nadir during the inactive phase[3,4]. The functional relevance of ultradian glucocorticoid pulses is still not fully understood; however, some ambitious studies have demonstrated that pulsatile activation is necessary to maintain optimal transcriptional activity of glucocorticoid receptors (GRs) in cell lines, and rat liver and brain[5–8], as well as HPA axis sensitivity, neuronal activation and behavioural responses to stress in rats[9]. In humans, an oscillating pattern of plasma cortisol has been demonstrated to be important for maintenance of healthy brain functions, including normal emotional and cognitive responses[10].

Feeding behaviour is another fundamental glucocorticoid-responsive activity[11], yet its regulation by physiological circadian and ultradian glucocorticoid rhythms has not been assessed despite its obvious importance for individuals whose natural rhythms are disrupted due to shift-work. To date, the basis of our understanding about feeding behaviour and glucocorticoids has been from seminal studies which used non-physiological glucocorticoid manipulations. For example, exogenous chronic CORT treatment in mice, delivered *ad libitum* in drinking water, resulted in high CORT levels during the daily active phase[12], reflecting increased drinking behaviour during this time, as previously shown[13]. These mice exhibited hyperphagia and obesity along with increased hypothalamic gene expression of *agouti-related neuropeptide* (*Agrp*), but without significant effect on *neuropeptide Y* (*Npy*), *proopiomelanocortin* (*Pomc*) or *cocaine and amphetamine regulated transcript* (*CART*) *prepropeptide* (*Cartpt*)[12]. AgRP and NPY are known for their role in promoting feeding behaviour, whereas POMC and CART are associated with satiety. In another early elegant study, chronic pharmacological level CORT exposure in adrenalectomized rats was induced experimentally by subcutaneously implanted CORT pellets, resulting in elevated *Npy* gene expression in the arcuate nucleus, as well as increased peptide synthesis, receptor activity, and feeding behaviour[14,15]. However, there was little impact of reduced CORT levels on *Npy* gene expression, peptide levels or receptor binding in adrenalectomized rats, leading to speculation that the rhythmic increase in CORT levels closely followed by a sharp rise in *Npy* expression at the start of the active phase may be the basis of the functional interaction.

In our current study we investigate the mechanisms underlying the adverse metabolic effects associated with phase shifting the circadian glucocorticoid profile. Using our model of physiological CORT replacement in adrenalectomized rats, we have assessed how feeding behaviour becomes dysregulated when the circulating CORT rhythm is out of phase with daily light:dark cues. To achieve this, our first experimental group was replaced with pulsatile CORT *in-phase* with light:dark cues, which is the most faithful representation of normal circadian and ultradian CORT exposure (Fig. 1a, c). The second group was replaced with the same pattern of CORT but exactly 12 h shifted *out-of-phase* with light:dark cues (Fig. 1b, d). The dose of CORT delivered was chosen to match average endogenous circulating CORT determined from our automated blood sampling system of adrenal-intact rats (S Fig. 1). Circadian activity and core body temperature profiles, timing and amount of food and water ingested, as well as gene expression of feeding regulating neuropeptides in the

hypothalamus and clock genes in the suprachiasmatic nucleus, were assessed in comparison to sham-operated (adrenal-intact) control rats, hereafter referred to as the control group.

To interrogate the glucocorticoid-dependence of the phenomenon, in the absence of time of day influence, we analysed dissected tissue containing the hypothalamic arcuate nucleus from adrenalectomized rats treated with an acute CORT injection at ZT2. Using chromatin immunoprecipitation (ChIP) with antibodies specific for GR, and the active initiating form of RNA Polymerase 2, phosphorylated on Serine 5 (pSer5 RNA-Pol2), we first confirmed that the well characterised CORT-responsive *Period1* (*Per1*) gene's glucocorticoid response element (GRE) and transcriptional start site (TSS) became GR bound and enriched with pSer5 RNA-Pol2 respectively, in a CORT-dependent manner, in the hypothalamic arcuate nucleus. Using the same chromatin preparations, we were able to identify CORT-dependent GR binding at a GRE in the 5'-regulatory region of *Npy*[16], along with a concomitant increase in pSer5 RNA-Pol2 occupancy at the TSS of the *Npy* gene. Taken together, these data support a direct glucocorticoid-dependent transcriptional regulation of *Npy* similarly to that previously shown for *Per1*. This supports our hypothesis that expression of *Npy* can become dysregulated during *out-of-phase* CORT exposure resulting in orexigenic behaviour independent of environmental daily cues.

## Results

**Timing of food intake is disordered when corticosterone is out-of-phase with daily light cues.** Surgery was carried out on Day 0. CORT infusions commenced on Day 1. Twelve-hourly food intake was measured until day 4. Total amount of food intake (g) per day was comparable between *in-phase*, *out-of-phase*, and sham-operated control groups during the experiment (Fig. 2a). However, the timing of food intake was significantly altered in the *out-of-phase group*. For the *in-phase* group, food intake was significantly lower during the light period (1.72 ± 1.27 g) than during the dark period (13.89 ± 2.55 g); comparable to the feeding behaviour of the sham-operated control group (light period, 1.94 ± 0.83 g; dark period, 13.50 ± 2.82 g). In contrast, the *out-of-phase* group ingested significantly more food during the light period (6.75 ± 1.87 g) compared to either *in-phase* or control groups. In fact, the amount of food ingested by the *out-of-phase* group during the light period was not significantly different to the amount they consumed during the dark period (7.83 ± 2.88 g) (Fig. 2b). Therefore, for the *out-of-phase* group, only 53.6% of food consumption occurred during the dark period (Fig. 2c), which was markedly different to both the *in-phase* and control groups that consumed most of their food during the dark period (89.1% and 88.4% respectively). Cumulative food intake analysis revealed a striking time of day pattern in food consumption for the *in-phase group*, which was ablated in the *out-of-phase* group (Fig. 2d). However, the timing of water intake was not altered in the *out-of-phase group* (S Fig. 2). Furthermore, body weight remained unchanged between both groups, and the *out-of-phase* CORT treatment had no impact on subcutaneous and epididymal fat mass (S Fig. 3).

**Daily timing of gene expression for the hypothalamic feeding regulating neuropeptides is reversed when corticosterone is out-of-phase with environmental daily light cues.** In situ hybridization histochemistry (ISHH) was performed on brains taken from rats in each of the three groups at either Zeitgeber Time (ZT) 1 or 13, on the final day of the experiment. Feeding regulating neuropeptides in the arcuate nucleus and lateral hypothalamic area were analysed (Fig. 3a). The gene expression of *Npy*, *Agrp*, *Pomc*, and *Cartpt* in the arcuate nucleus and

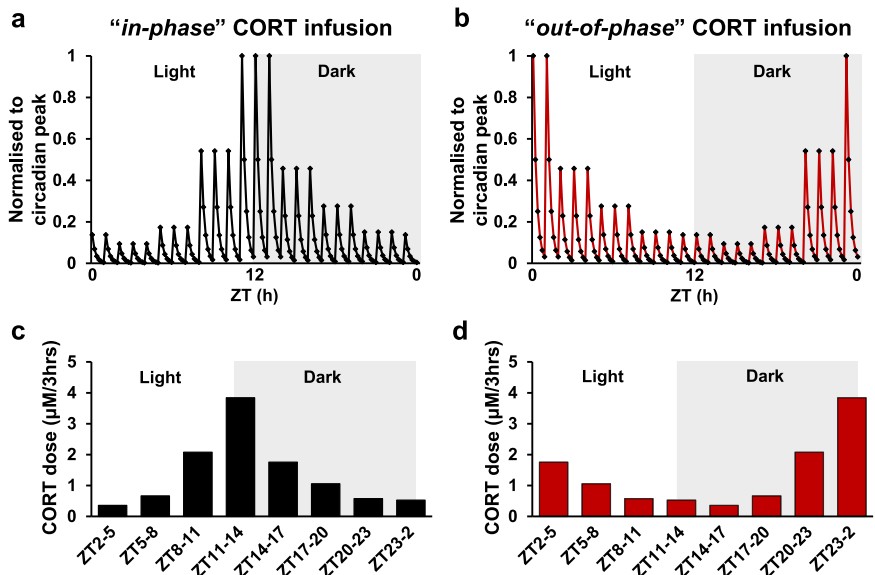

**Fig. 1 Schematic of circadian and ultradian CORT infusion profiles.** Schematic representation of circadian and ultradian CORT infusion pattern *in-phase* (**a**) or *out-of-phase* (**b**). Dose of CORT (μM/3 h) for *in-phase* (**c**) and *out-of-phase* (**d**) infusions. The total amount of CORT given per day was equivalent between the two groups.

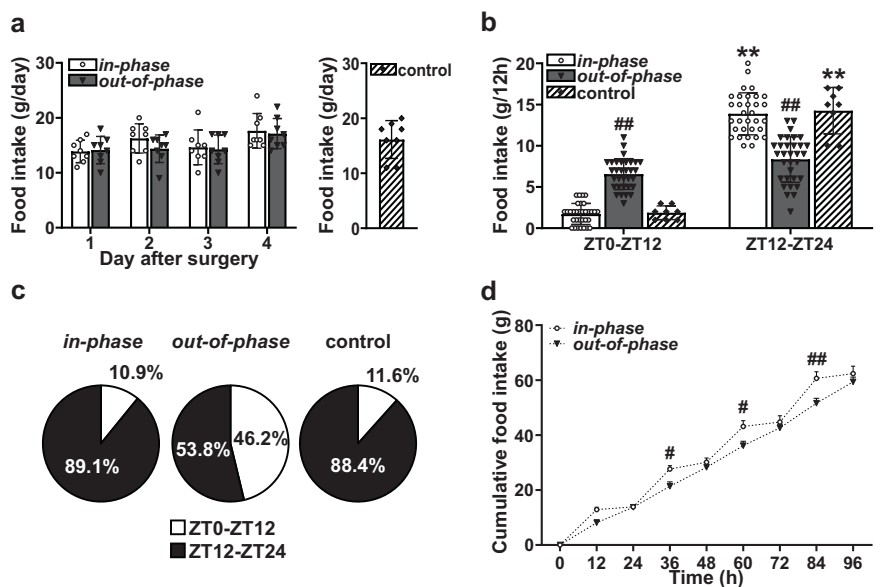

**Fig. 2 Dysregulated feeding pattern during out-of-phase glucocorticoid exposure. a** Total food intake (g/day) over the course of the experiment; Repeated Measures 2-Way ANOVA ($p = 0.47$, infusion pattern; $p < 0.01$, day of treatment, $p = 0.66$, interaction between infusion pattern and day of treatment). **b** Twelve-hourly food intake was measured until day 4. Significant infusion pattern-dependent effects on timing of food intake (g/12 h); Repeated Measures 2-Way ANOVA ($p = 0.47$, infusion pattern; $p < 0.01$, time of day; $p < 0.01$, interaction between infusion pattern and time of day). **c** Percentage of food intake during light period (ZT0–ZT12) and dark period (ZT12–ZT24) for *in-phase*, *out-of-phase*, and control groups. **d** Cumulative measurements throughout the experiment showed a distinctive step-wise increment in pattern of food intake for *in-phase* group but not *out-of-phase* group. Repeated Measures 2-Way ANOVA ($p < 0.01$, infusion pattern; $p < 0.01$, time of day; $p < 0.01$, interaction between infusion pattern and time of day). Data are presented as mean ± SD, with individual datapoints representing biological repeats shown on each graph. Bonferroni multiple comparison post-test results are shown on the graphs; Significant one-to-one differences between timepoints (within the same treatment) indicated by *$p < 0.05$, **$p < 0.01$ and between treatment (at the same timepoint) indicated by #$p < 0.05$, ##$p < 0.01$.

*pro-melanin-concentrating-hormone* (*Pmch*) in the lateral hypothalamic area were measured (Fig. 3b). The captured photographs are featured within supplementary materials (S Fig. 4). *Npy*, *Agrp*, and *Pmch*, which are known orexigenic neuropeptides, were significantly increased at ZT13 in both the *in-phase* and control groups (*Npy*, *in-phase* 1.83 ± 0.31, control 1.74 ± 0.30; *Agrp*, *in-phase* 1.69 ± 0.38, control 1.56 ± 0.22; *Pmch*, *in-phase* 1.62 ± 0.26, control 1.46 ± 0.21) compared to ZT1 (*Npy*, *in-phase* 1.13 ± 0.12, control 1.00 ± 0.18; *Agrp*, in-phase 1.05 ± 0.27, control 1.00 ± 0.21; *Pmch*, *in-phase* 1.03 ± 0.20, control 1.00 ± 0.17) (Fig. 3c). In stark contrast, the orexigenic gene expression pattern was reversed in the *out-of-phase* group, with all significantly decreased at ZT13 (*Npy*, 1.11 ± 0.32; *Agrp*, 1.12 ± 0.11; *Pmch*, 1.03 ± 0.16) compared to ZT1 (*Npy*, 1.78 ± 0.47; *Agrp*, 1.56 ± 0.22; *Pmch*, 1.35 ± 0.21)

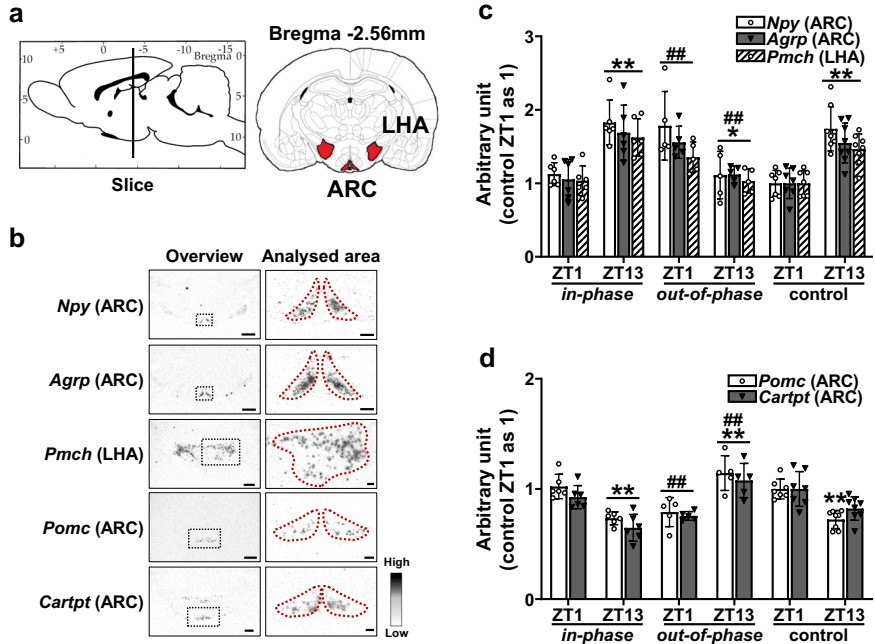

**Fig. 3 Altered expression of hypothalamic feeding regulating neuropeptide genes during out-of-phase glucocorticoid exposure. a** Schematic showing analysed brain regions of the arcuate nucleus (ARC) and lateral hypothalamic area (LHA) (schematic adapted from image taken from rat brain atlas[67]). **b** Representative digital images of ISHH of *Npy*, *Agrp*, *Pmch*, *Pomc* and *Cartpt*. Low magnification overview image with the area to be analysed indicated by the black dotted line, and higher magnification image with analysed area indicated by red dotted line. Scale bars indicate 500 μm (low magnification overview image) and 200 μm (higher magnification image). **c** Significant infusion pattern-dependent effects were found for the timing of gene expression for orexigenic peptides *Npy*, *Agrp* and *Pmch*; 2-Way ANOVA (*Npy*: $p < 0.05$, infusion pattern; $p = 0.64$, time of day; $p < 0.01$, interaction; *Agrp*: $p < 0.01$, infusion pattern; $p = 0.62$, time of day; $p < 0.01$, interaction; *Pmch*: $p < 0.01$, infusion pattern; $p = 0.28$, time of day; $p < 0.01$, interaction). **d** Significant infusion pattern-dependent effects were found for the timing of gene expression for anorexigenic peptides *Pomc* and *Cartpt*; 2-Way ANOVA (*Pomc*: $p < 0.05$, infusion pattern; $p = 0.05$, time of day; $p < 0.01$ interaction; *Cartpt*: $p < 0.05$, infusion pattern; $p = 0.28$, time of day; $p < 0.01$, interaction). Data are presented as mean ± SD, with individual datapoints representing biological repeats shown on each graph. Bonferroni multiple comparison post-test results are shown on the graphs; Significant one-to-one differences between timepoints (within the same treatment) indicated by *$p < 0.05$, **$p < 0.01$ and between treatment (at the same timepoint) indicated by #$p < 0.05$, ##$p < 0.01$.

(Fig. 3c). Compared to the orexigenic neuropeptides, an expected opposing expression pattern was evident for the anorexigenic *Pomc* and *Cartpt*. These were significantly decreased at ZT13 for both the *in-phase* and control groups (*Pomc*, *in-phase* 0.73 ± 0.06, control, 0.72 ± 0.08; *Cartpt*, *in-phase* 0.65 ± 0.12, control, 0.82 ± 0.11) compared to ZT1 (*Pomc*, *in-phase* 1.02 ± 0.11, control 1.00 ± 0.09; *Cartpt*, *in-phase* 0.93 ± 0.10, control 1.00 ± 0.16) (Fig. 3d). However, the reverse was seen for the *out-of-phase* group. *Pomc* and *Cartpt* were markedly increased at ZT13 (*Pomc*, 1.14 ± 0.16; *Cartpt*, 1.08 ± 0.15) in comparison to ZT1 (*Pomc*, 0.79 ± 0.13; *Cartpt*, 0.75 ± 0.04) (Fig. 3d).

**Effects of out-of-phase corticosterone exposure on HPA axis, oxytocin (*Oxt*), and arginine vasopressin (*Avp*).** The effect of *out-of-phase* CORT exposure on the HPA axis was next assessed. Gene expression of *corticotrophin releasing hormone* (*Crh*) in the paraventricular nucleus and *Pomc* in the anterior pituitary were measured by ISHH (Fig. 4a, b). For both *in-phase* and control groups, *Crh* and *Pomc* were significantly higher at ZT13 (*Crh*, *in-phase* 1.61 ± 0.41, control 1.57 ± 0.24; *Pomc*, *in-phase* 2.83 ± 0.70, control 2.49 ± 0.38) compared to ZT1 (*Crh*, *in-phase* 0.99 ± 0.25, control 1.00 ± 0.18; *Pomc*, *in-phase* 1.60 ± 0.20, control 1.00 ± 0.34) in each group (Fig. 4c). Whilst they were also increased in the *out-of-phase* group at ZT13 (*Crh*, 1.85 ± 0.26; *Pomc*, 4.09 ± 0.72) compared to ZT1 (*Crh*, 1.07 ± 0.21; *Pomc*, 2.34 ± 0.67); increased *Pomc* expression at ZT13 in *out-of-phase* was markedly elevated (Fig. 4c). Plasma adrenocorticotrophic hormone (ACTH) was markedly increased at ZT13, for both

*in-phase* (652.9 ± 543.9 pg/mL) and *out-of-phase* (1808.3 ± 559.6 pg/mL) groups, in comparison to ZT1 (*in-phase*, 92.6 ± 38.6 pg/mL; *out-of-phase*, 148.4 ± 207.4 pg/mL); however ACTH at ZT13 was much higher in the *out-of-phase* than the *in-phase* group (Fig. 4d) consistent with the results of ISHH. As with the expected CORT profiles from our programmed infusion pump settings, plasma CORT concentration was significantly higher at ZT13 for the *in-phase* (256.1 ± 63.8 ng/mL) and at ZT1 for the *out-of-phase* (391.2 ± 83.5 ng/mL) groups, compared to ZT1 for the *in-phase* (103.1 ± 75.3 ng/mL) and ZT13 for the *out-of-phase* (161.7 ± 97.0 ng/mL) groups, respectively (Fig. 4e).

*Oxt* and *Avp* gene expression in the supraoptic nucleus and paraventricular nucleus were also analysed by ISHH (Fig. 5a). They were both significantly decreased in both *in-phase* and control groups at ZT13 (*Oxt* in the supraoptic nucleus, *in-phase* 0.66 ± 0.15, control 0.58 ± 0.12; *Oxt* in the paraventricular nucleus, *in-phase* 0.67 ± 0.16, control 0.75 ± 0.15; *Avp* in the supraoptic nucleus, *in-phase* 0.62 ± 0.23, control 0.74 ± 0.19; *Avp* in the paraventricular nucleus, *in-phase* 0.76 ± 0.24, control 0.63 ± 0.27) compared to ZT1 (*Oxt* in the supraoptic nucleus, *in-phase* 0.95 ± 0.13, control 1.00 ± 0.14; *Oxt* in the paraventricular nucleus, *in-phase* 0.99 ± 0.12, control 1.00 ± 0.19; *Avp* in the supraoptic nucleus, *in-phase* 1.04 ± 0.13, control 1.00 ± 0.21; *Avp* in the paraventricular nucleus, *in-phase* 1.03 ± 0.32, control 1.00 ± 0.30) (Fig. 5b). Similar to our findings with the orexigenic and anorexigenic neuropeptides, these expression patterns were reversed in the *out-of-phase* group, with significantly increased expression at ZT13 (*Oxt* in the supraoptic nucleus, 0.89 ± 0.11;

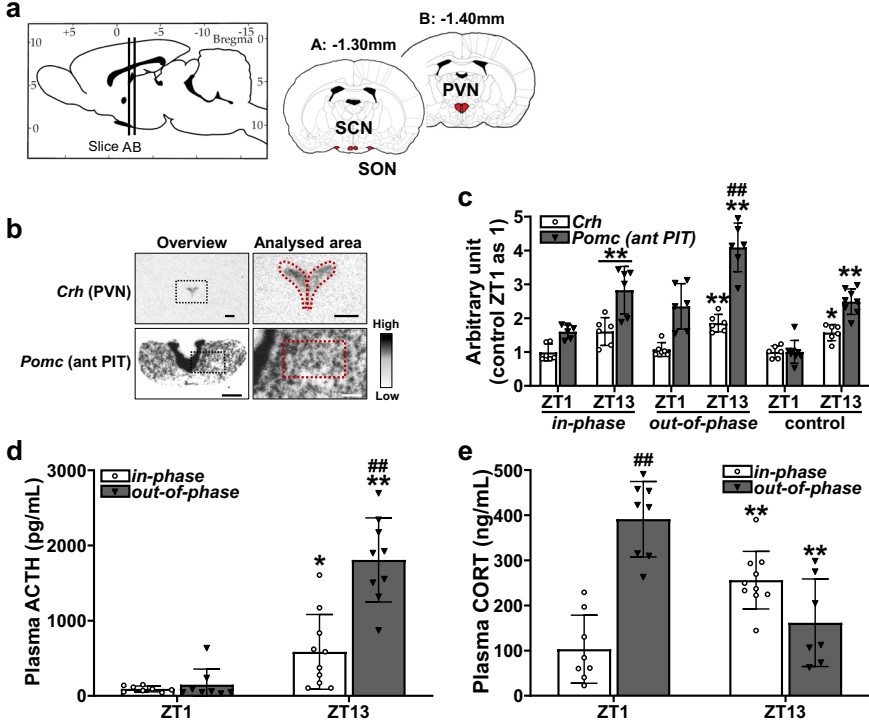

**Fig. 4 Differential gene expression pattern of HPA axis regulators *Crh* and *Pomc* between in-phase and out-of-phase CORT exposure. a** Schematic showing analysed brain regions of the suprachiasmatic nucleus (SCN), supraoptic nucleus (SON), and paraventricular nucleus (PVN) (schematic adapted from image taken from rat brain atlas[67]). **b** Representative digital images of ISHH of *Crh* in the PVN and *Pomc* in the anterior pituitary (ant PIT). Low magnification overview image with the area to be analysed indicated by the black dotted line, and higher magnification image with analysed area indicated by red dotted line. Scale bars indicate 500 μm (low magnification overview image) and 200 μm (higher magnification image). **c** Significant infusion pattern-dependent effects were found for the timing of *Crh* and *Pomc* gene expression; 2-Way ANOVA (*Crh*: $p < 0.01$, infusion pattern; $p = 0.21$, time of day; $p = 0.61$; interaction; *Pomc*: $p < 0.01$, infusion pattern; $p < 0.01$, time of day; $p = 0.49$; interaction). **d** Plasma adrenocorticotrophic hormone (ACTH, pg/mL) were measured from trunk blood samples at the end of the experiment. Significant infusion pattern-dependent effects were found for plasma ACTH concentration; 2-Way ANOVA ($p < 0.01$, infusion pattern; $p < 0.01$, time of day; $p < 0.01$, interaction). **e** Plasma corticosterone (CORT, ng/mL) were measured from the trunk blood samples at the end of the experiment. Significant infusion pattern-dependent effects were found for plasma CORT concentration; 2-Way ANOVA ($p < 0.01$, infusion pattern; $p = 0.18$, time of day; $p < 0.01$, interaction). Data are presented as mean ± SD, with individual datapoints representing biological repeats shown on each graph. Bonferroni multiple comparison post-test results are shown on the graphs; Significant one-to-one differences between timepoints (within the same treatment) indicated by *$p < 0.05$, **$p < 0.01$ and between treatment (at the same timepoint) indicated by #$p < 0.05$, ##$p < 0.01$.

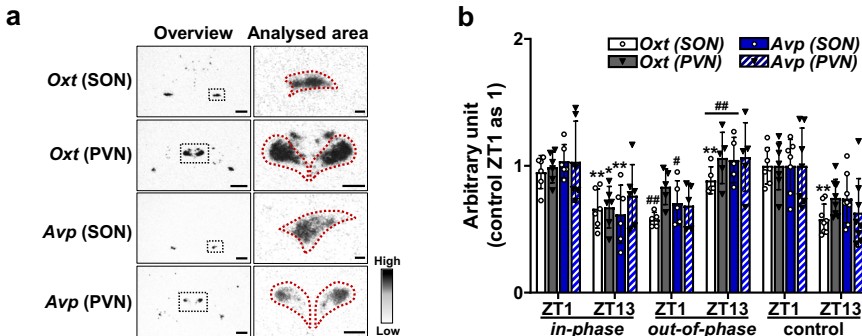

**Fig. 5 Differential gene expression pattern of *Oxt* and *Avp* between in-phase and out-of-phase CORT exposure. a** Representative digital images of ISHH of *oxytocin* (*Oxt*) and *arginine vasopressin* (*Avp*) in the supraoptic nucleus (SON) and paraventricular nucleus (PVN). Low magnification overview image with the area to be analysis indicated by the black dotted line, and higher magnification image with analysed area indicated by red dotted line. Scale bars indicate 500 μm (low magnification overview image) and 200 μm (higher magnification image). **b** Significant infusion pattern-dependent effects were found for the timing of gene expression for *Oxt* and *Avp*; 2-Way ANOVA (*Oxt* (SON): $p < 0.01$, infusion pattern; $p = 0.35$, time of day; $p < 0.01$, interaction; *Oxt* (PVN): $p < 0.05$, infusion pattern; $p = 0.25$, time of day; $p < 0.01$, interaction; *Avp* (SON): $p = 0.09$, infusion pattern; $p = 0.79$, time of day; $p < 0.01$, interaction; *Avp* (PVN): $p = 0.35$, infusion pattern; $p = 0.71$, time of day; $p < 0.01$, interaction). Data are presented as mean ± SD, with individual datapoints representing biological repeats shown on each graph. Bonferroni multiple comparison post-test results are shown on the graphs; Significant one-to-one differences between timepoints (within the same treatment) indicated by *$p < 0.05$, **$p < 0.01$ and between treatment (at the same timepoint) indicated by #$p < 0.05$, ##$p < 0.01$.

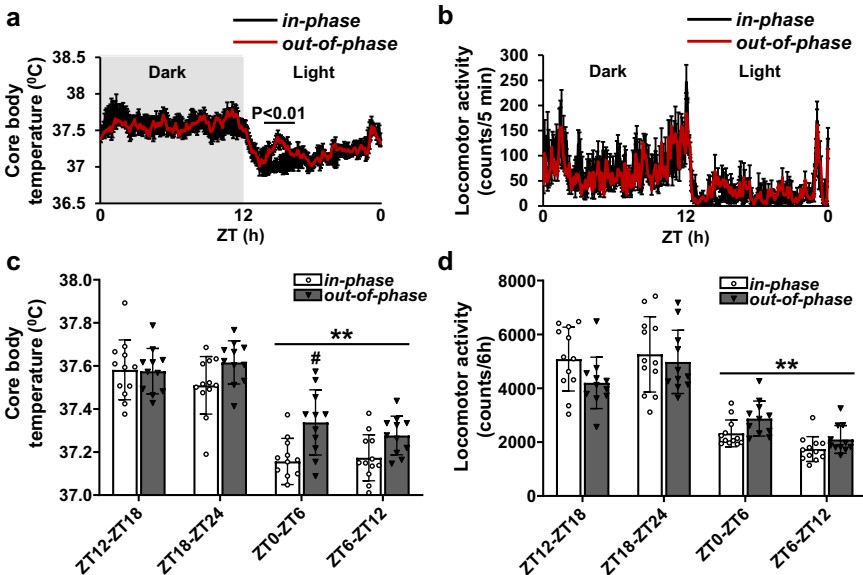

**Fig. 6 Effects of dysregulated glucocorticoid exposure on core body temperature, but not locomotor activity. a** Averaged locomotor activity for in-phase and out-of-phase groups over 24 h. **b** Averaged core body temperature (°C) for in-phase and out-of-phase groups over 24 h. **c** Total locomotor activity in 6 h epochs were compared for in-phase and out-of-phase groups. Repeated Measures 2-Way ANOVA ($p = 0.73$, infusion pattern; $p < 0.01$, time of day; $p < 0.05$, interaction). **d** Averaged core body temperature (°C) in 6 h epochs were compared for in-phase and out-of-phase groups. Repeated Measures 2-Way ANOVA $p < 0.01$, infusion pattern; $p < 0.01$, time of day; $p = 0.07$, interaction Data are presented as mean ± SD, with individual datapoints representing biological repeats shown on each graph. Bonferroni multiple comparison post-test results are shown on the graphs; Significant one-to-one differences between timepoints (within the same treatment) indicated by *$p < 0.05$, **$p < 0.01$ and between treatment (at the same timepoint) indicated by #$p < 0.05$, ##$p < 0.01$.

*Oxt* in the paraventricular nucleus, $1.06 \pm 0.20$; *Avp* in the supraoptic nucleus, $1.05 \pm 0.18$; *Avp* in the paraventricular nucleus, $1.07 \pm 0.27$) compared to ZT1 (*Oxt* in the supraoptic nucleus, $0.58 \pm 0.06$; *Oxt* in the paraventricular nucleus, $0.83 \pm 0.14$; *Avp* in the supraoptic nucleus, $0.71 \pm 0.18$; *Avp* in the paraventricular nucleus, $0.69 \pm 0.18$) (Fig. 5b). The captured photographs are featured within supplementary materials (S Fig. 5).

**Circadian core body temperature (CBT), activity, and clock genes**. Significant differences in CBT between the *in-phase and out-of-phase* groups were restricted to the early part of the active phase. The *out-of-phase* group had higher CBT at ZT0-ZT6 ($37.34 \pm 0.15\,°C$) compared to *in-phase* ($37.16 \pm 0.11\,°C$) (Fig. 6a, c). There were no statistically significant differences between *in-phase* and *out-of-phase* groups at ZT12-ZT18 (*in-phase* $37.58 \pm 0.14\,°C$, *out-of-phase* $37.58 \pm 0.11\,°C$), ZT18-ZT24 (*in-phase* $37.51 \pm 0.13\,°C$, *out-of-phase* $37.62 \pm 0.10\,°C$), and ZT6-ZT12 (*in-phase* $37.17 \pm 0.11\,°C$, *out-of-phase* $37.28 \pm 0.09\,°C$) (Fig. 6c). Overall, mean core body temperature during the light period was still markedly lower compared to mean core body temperature during the dark period (Fig. 6c) irrespective of group.

As the elevated CBT levels were restricted to the highest amplitude CORT pulses at the start of the inactive phase, we speculated that this may be due to a direct CORT effect rather than a simple post-prandial hyperthermic effect. To test this, CBT was monitored over a 6 h fasting period from ZT23 to ZT5. At ZT1, either VEH (HBC-Saline) or CORT-HBC-Saline (3 mg/kg) was injected subcutaneously. CBT rose sharply in the CORT treated rats, and remained elevated for approximately 2 h after the injection (S Fig. 6). No effect on CBT was found for the HBC-Saline treated rats.

In contrast to CBT, locomotor activity was not affected by *out-of-phase* CORT infusion (Fig. 6b, d). No significant difference was evident in locomotor activity, assessed in 6 h epochs, between *in-phase* and *out-of-phase* groups (ZT12-ZT18, *in-phase* $5085.2 \pm 1185.3$, *out-of-phase* $4201.4 \pm 956.5$; ZT18-ZT24, *in-phase* $5260.0 \pm 1400.2$, *out-of-phase* $4980.3 \pm 1174.4$; ZT0-ZT6, *in-phase* $2325.6 \pm 498.8$, *out-of-phase* $2874.8 \pm 650.3$; ZT6-ZT12, *in-phase* $1741.8 \pm 462.7$, *out-of-phase* $2098.7 \pm 517.8$). As expected, locomotor activity was significantly lower during the light period than during the dark period in both groups (Fig. 6d).

*Per1*, *Per2*, and heteronuclear (hn) *Avp* gene expression in the suprachiasmatic nucleus and *Hcrt* in the lateral hypothalamic area were analysed by ISHH (Fig. 7a). For all groups, *Per1* and *hnAvp* expression were significantly decreased at ZT13 (*Per1*, *in-phase*, $0.41 \pm 0.05$, *out-of-phase* $0.35 \pm 0.03$, control, $0.46 \pm 0.04$; *hnAvp*, *in-phase*, $0.27 \pm 0.09$, *out-of-phase* $0.33 \pm 0.09$, control, $0.28 \pm 0.06$) compared to ZT1 (*Per1*, *in-phase*, $0.95 \pm 0.14$, *out-of-phase* $0.91 \pm 0.06$, control, $1.00 \pm 0.09$; *hnAvp*, *in-phase*, $1.15 \pm 0.28$, *out-of-phase* $1.01 \pm 0.15$, control, $1.00 \pm 0.15$) (Fig. 7b). For all groups, *Per2* expression was markedly increased at ZT13 (*in-phase*, $5.44 \pm 2.13$, *out-of-phase* $4.26 \pm 1.36$, control, $5.85 \pm 1.30$) in comparison to ZT1 (*in-phase*, $0.74 \pm 0.32$, *out-of-phase* $0.75 \pm 0.25$, control, $1.00 \pm 0.18$) in all groups (Fig. 7c). *Hcrt* was dramatically increased at ZT13 (*in-phase*, $2.33 \pm 0.26$, *out-of-phase* $2.15 \pm 0.31$, control, $2.20 \pm 0.23$) compared to ZT1 (*in-phase*, $1.02 \pm 0.11$, *out-of-phase* $1.03 \pm 0.13$, control, $1.00 \pm 0.19$) in all groups (Fig. 7d). Taken together, these data indicate that *clock* genes in the suprachiasmatic nucleus and *Hcrt* in the lateral hypothalamic area remain entrained by light/dark cues and are not disrupted by *out-of-phase* CORT exposure. The captured photographs are featured within supplementary materials (S Fig. 7).

**CORT-induced GR binding at GRE sites and pSer5 RNA Pol2 enrichment at TSS of *Per1* and *Npy* genes in the arcuate nucleus**. Finally, to determine whether CORT exposure is necessary and sufficient to induce increased transcription rate of

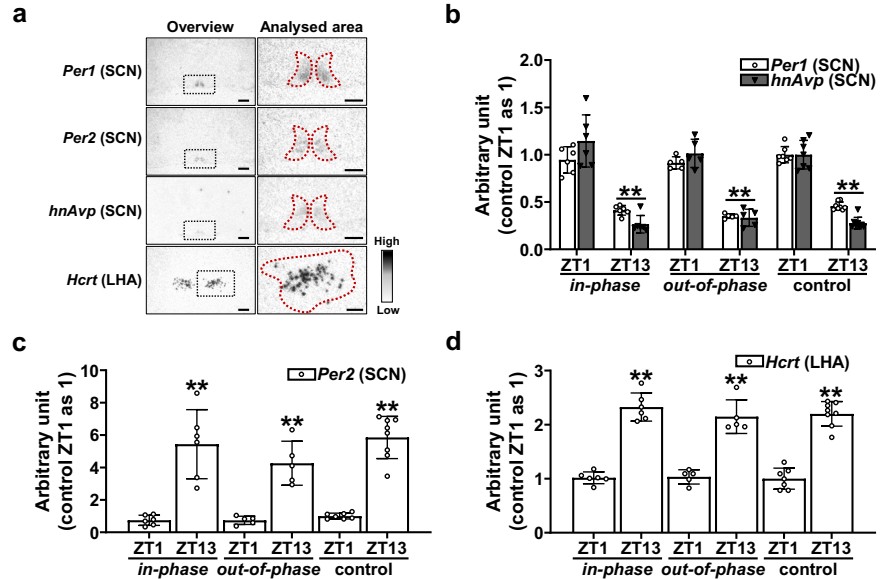

**Fig. 7 Effects of dysregulated glucocorticoid exposure on Clock-regulating genes and *Hypocretin neuropeptide precursor Orexin*. a** Representative digital images of ISHH of *Period 1* (*Per1*), *Period 2* (*Per2*), and *heteronuclear arginine vasopressin* (*hnAvp*) in the suprachiasmatic nucleus (SCN) and *Hypocretin neuropeptide precursor* (*Hcrt*; previously known as *Orexin*) in the lateral hypothalamic area (LHA). Low magnification overview image with the area to be analysis indicated by the black dotted line, and higher magnification image with analysed area indicated by red dotted line. Scale bars indicate 500 μm (low magnification overview image) and 200 μm (higher magnification image). **b** Regulation of SCN *Per1* and *hnAvp*; 2-Way ANOVA (Per1: $p < 0.05$, infusion pattern; $p < 0.01$, time of day; $p = 0.90$, interaction; *hnAvp*: $p = 0.52$, infusion pattern; $p < 0.01$, time of day; $p = 0.26$, interaction). **c** Regulation of SCN *Per2*; 2-Way ANOVA ($p = 0.17$, infusion pattern; $p < 0.01$, time of day; $p = 0.36$, interaction). **d** Regulation of *Hcrt*; 2-Way ANOVA ($p = 0.61$, infusion pattern; $p < 0.01$, time of day; $p = 0.57$, interaction). Data are presented as mean ± SD, with individual datapoints representing biological repeats shown on each graph. Bonferroni multiple comparison post-test results are shown on the graphs; Significant one-to-one differences between timepoints (within the same treatment) indicated by *$p < 0.05$, **$p < 0.01$.

the *Npy* gene, we performed a reductionist experiment to test the effect of an acute CORT treatment at ZT2. Adrenalectomized rats were treated with an acute subcutaneous injection of either CORT (3 mg/kg)-HBC-Saline or HBC-Saline (Vehicle control) at ZT1, then exactly one hour later at ZT2, the rats were euthanized and hypothalamic arcuate nucleus regions dissected and fixed for ChIP assay (Fig. 8a). As expected, plasma CORT concentrations were extremely low in adrenalectomized rats, and consistently elevated at 1 h after CORT injection (Fig. 8b). As a positive control to confirm CORT-responsiveness within the arcuate nucleus, GR binding at the hypersensitive *Per1* GRE was first tested. The acute CORT treatment was seen to induce a robust increase of $15.58 ± 3.07$ fold change in GR detected at the *Per1* GRE (Fig. 8c). Unpaired *t* test indicated the change to be statistically significant ($p = 0.0019$). A concomitant robust increase of $16.63 ± 2.49$ fold change in pSer5 RNA Pol2 enrichment at *Per1* TSS was similarly statistically significant ($p = 0.0010$) (Fig. 8d). For GR binding at a relatively less well-characterized GRE, which has been described at a site upstream of the *Npy* gene[16], a fold change of $6.77 ± 1.32$ in CORT-inducibility was also found to be statistically significant ($p = 0.0015$) (Fig. 8e). A concomitant increase of $3.82 ± 1.13$ fold change in pSer5 RNA Pol2 enrichment at *Npy* TSS was also found to be statistically significant ($p = 0.0106$) (Fig. 8f) indicating that acute CORT exposure at ZT2 was necessary and sufficient to induce GR binding at the *Npy* gene's upstream GRE and increased pSer5 Pol2 occupancy at the *Npy* gene's TSS.

## Discussion

We have demonstrated that desynchronizing circulating CORT rhythms from daily light cues has a significant impact on feeding behaviour. The timing of food intake was significantly affected when CORT exposure was *out-of-phase* with light:dark cues,

resulting in a profound shift in time of eating. Notably, we found increased appetitive behaviour during the inactive phase of the day in the *out-of-phase* group. The altered profile in food intake was associated with aberrant timing in the regulation of orexigenic and anorexigenic neuropeptides in the hypothalamus. Circadian activity profiles remained entrained by daily light:dark cues, along with maintenance of entrained oscillations of *clock* genes in the suprachiasmatic nucleus and *Hcrt* in the lateral hypothalamic area, irrespective of CORT rhythm. To our knowledge, this study provides the first evidence that synchronization of environmental daily light:dark cues with the 'endogenous' circulating CORT profile in rats is essential for maintaining appetitive behaviour for increased food consumption during the active phase and decreased food consumption during the inactive phase.

Although food intake is often considered a homoeostatic behaviour, there is less understanding how food intake may act as an automatic response to an acute shortage of energy. As food intake mainly occurs during the dark period in nocturnal animals when under *ad libitum* fed conditions, onset of the 'lights off' signal is considered as one of the possible major factors that drives food intake[17]. However, the amount of food consumption is generally determined by numerous hormones and neuropeptides which are secreted both in the central nervous system and other peripheral organs in response to digested nutrients[18–20], and, importantly, other factors such as stress, learning, and palatability, which all affect the amount and timing of food intake[21].

In the present study, the total amount of food consumed per day was similar between the *in-phase*, *out-of-phase*, and sham-operated control groups. Importantly for our study design, the relative food intake during the active versus inactive periods was comparable between *in-phase* and control groups, strongly

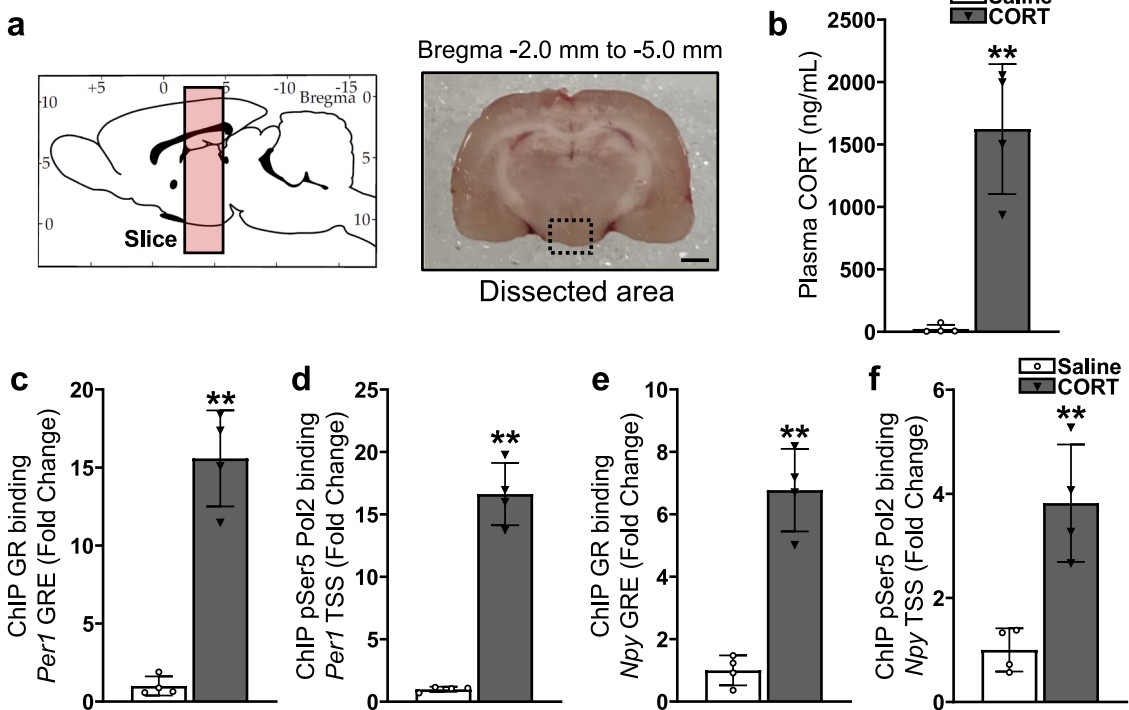

**Fig. 8 CORT-induced GR binding at GRE sites and pSer5 RNA Pol2 enrichment at TSS of *Per1* and *Npy* genes in the arcuate nucleus. a** Region dissected from hypothalamus (co-ordinates bregma −2.0 mm to −5.0 mm, schematic adapted from image taken from rat brain atlas[67]) used for collection of arcuate nucleus-enriched tissue (photograph by author) for ChIP processing. Scale bar indicates 2 mm. **b** CORT concentration in trunk blood collected from adrenalectomized rats at exactly 1 h after acute subcutaneous injection of either CORT (3 mg/kg)-HBC-Saline or HBC-Saline (Vehicle control). Significantly higher CORT levels were detected in CORT-injected compared to vehicle-injected controls (**$p = 0.0085$). **c** GR binding at the hypersensitive *Per1* GRE was significantly induced by the acute CORT injection (**$p = 0.0019$). **d** pSer5 RNA Pol2 enrichment at *Per1* TSS was significantly increased by the acute CORT injection (**$p = 0.0010$). **e** GR binding at the GRE upstream of the *Npy* gene was significantly induced by the acute CORT injection (**$p = 0.0015$). **f** pSer5 RNA Pol2 enrichment at *Npy* TSS was significantly increased by the acute CORT injection (*$p = 0.0106$). Data are presented as mean ± SD ($n = 4$ each).

indicating that our programmed CORT infusion pattern, which was modelled on adrenal-intact rats, faithfully recapitulated a physiologically realistic endogenous CORT secretion pattern. Interestingly, rats in the *out-of-phase* group ate continuously, even during the inactive period. Although CORT exposure was expected to induce feeding[14], food intake still occurred during the active phase, potentially indicating that when daily light cues are misaligned to CORT levels there is a continued drive to eat throughout the inactive as well as the active phase.

The results of ISHH, with regard to the feeding regulating neuropeptides in the hypothalamus, were also consistent with our hypothesis based on previous work. The orexigenic neuropeptide transcripts (*Npy*, *Agrp* and *Pmch*) were significantly up-regulated at ZT13 in both *in-phase* and control groups, as they ate much more during the dark period, ZT12–ZT24, compared to ZT0–ZT12. However, in the *out-of-phase* group, the observed reversed expression pattern of *Npy*, *Agrp* and *Pmch* would most certainly be the cause of decreased food consumption during dark period. For the anorexigenic neuropeptide transcripts (*Pomc* and *Cart*), food consumption would be expected to increase when their expression decreased. A previous study has shown that CORT directly binds to glucocorticoid type II receptors (GRs) expressed by NPY producing neurons to directly induce *Npy* gene expression, peptide synthesis, receptor activity, and NPY-induced feeding[14]. There is also abundant expression of GRs in the arcuate nucleus and lateral hypothalamic area. However no co-localization was observed in melanin-concentrating hormone neurons[22]. Thus, CORT most likely influences NPY, AgRP, POMC, and CART neurons by direct

actions, whereas gene expression of *Pmch* is more likely influenced indirectly via a negative feedback mechanism of the positive or negative energy balance.

At the cellular level, two major subsets of arcuate nucleus neurons that co-express NPY and AgRP were found to have abundant GR expression and exhibit electrophysiological and transcriptional CORT responsiveness[23–25]. Furthermore, an ambitious study by Leon-Mercado *et al*[26] assessing a novel function of the arcuate nucleus in negative feedback of the HPA axis, used local arcuate nucleus microdialysis with mineralocorticoid receptor (MR) and GR antagonists during either the circadian nadir or circadian peak. They found that arcuate nucleus sensitivity to CORT stimulation was GR dominant during the active time of day, consistent with the notion of a principle role for GR in these AgRP/NPY neurons. While functionally distinct from the regulation of feeding behaviour assessed in our study, the findings of Leon-Mercado et al[26] are consistent with a primary role for GR in AgRP/NPY circadian regulation of appetitive behaviour.

Interestingly, we also found *Agrp* expression was significantly altered in the *out-of-phase* treatment group. The regulation of *Agrp* is known to be more complex than that of *Npy*, with the fasting-induced brain-specific homeobox factor Bsx required to act together with GR to induce *Agrp* transcription[27]. As feeding behaviour is altered with out-of-phase CORT exposure, any manipulation of fed/fasted state required to assess *Agrp* regulation would introduce a major confounding factor. Therefore, we have been unable to interrogate GR-specific regulation of *Agrp* in the context of our experimental question.

Our data clearly demonstrates corticosterone-inducible GR binding at a GRE site in the *Npy* promoter and concomitant increase in enrichment of pSer5 RNA Pol2 at the *Npy* TSS. Although there is data from the 'dimerization mutant' GRdim mouse[28] suggesting Npy regulation may not need GR[27], more recent data has revealed that the GRdim model is in fact functional in certain contexts[29–32], which is further supported by our current data.

This evidence for GR's involvement in the transcriptional regulation of *Npy* in the arcuate nucleus does not preclude an additional involvement of the MR since both GR and MR are highly expressed in the arcuate nucleus[33–35]. However, because MR has 10 fold higher affinity for CORT than GR it is near maximally bound even at low CORT levels circulating during the circadian nadir, while in contrast the lower affinity GR becomes activated with the higher CORT levels of an ultradian pulse peak within the circadian active phase[36–39] and therefore provides a more responsive and dynamic sensor of fluctuating CORT levels over the day. Accordingly, GR is generally thought to be more likely to initiate transcriptional responses to higher CORT levels. Therefore, while it seems unlikely that the changes in appetitive behaviour are solely MR-mediated, it is possible that MR is involved in augmenting GR's transcriptional action on *Npy*, by acting in complex with GR in a manner similar to that described by Rivers et al[40].

With regards to the physiological data, drinking profiles were similar among the groups, even for the *out-of-phase* group (S Fig. 2). It was expected that the fluid intake profile would be altered along with the food intake profile in the *out-of-phase* group, as rats normally drink more when they eat; an important physiological adaptation to maintain the integrity of the fluid compartment of the body and balance osmolytes from food[41,42]. An experimental limitation to consider here, is that all adrenalectomized rats in the study were provided with 0.9% physiological saline instead of tap water for drinking, which is a requirement after adrenalectomy to maintain salt balance in the absence of adrenal mineralocorticoids[43,44]. Although we cannot explain the dissociation of the two closely related behaviours of increased eating without increased drinking, we can conclude that CORT was involved in feeding but not in drinking behaviour in this study design at least.

It should be noted that no change in body weight, subcutaneous fat mass nor epididymal fat mass was observed (S Fig. 3), presumably because the 5-day time course of our study was too brief to see any difference in these metabolic parameters. A much longer experimental time course, potentially along with the choice of a more palatable high fat/high sugar diet would be required to see changes in parameters such as body weight and fat deposition. Nevertheless, the remarkably robust changes in feeding behaviour were evident immediately and remained consistent throughout each of the five days of the experiment.

The AM/PM differences found by ISHH of *Crh* in the paraventricular nucleus and *Pomc* in the anterior pituitary indicate that their circadian rhythm remained intact in all groups, which is consistent with the expected pattern from previous reports[45]. However, *Crh* and *Pomc* mRNA expression, as well as circulating ACTH levels in the *out-of-phase* group were significantly elevated at ZT13 compared to the both *in-phase* and control groups at ZT13. Likely reflecting the lack of CORT-dependent negative feedback[46] at that time in the *out-of-phase* group.

The gene expression of *Oxt* and *Avp* exhibit AM/PM differences, as well as other feeding regulating neuropeptides in the hypothalamus[47–49]. These molecules are regarded as anorexigenic peptides as many studies have demonstrated that OXT and AVP decrease food intake[50,51]. The variation in expression of these genes was similar to those of anorexigenic neuropeptides in the arcuate nucleus. In addition, Itoi *et al.* have reported that gene expression of *Oxt* and *Avp* in the paraventricular nucleus were significantly downregulated by CORT exposure[52], supporting our results.

Strikingly, the rats' circadian activity profile remained normal even with *out-of-phase* CORT infusion. Circadian activity is mainly regulated by the master circadian clock in the suprachiasmatic nucleus as has been previously described[53]. The suprachiasmatic nucleus has been reported to be relatively devoid of GR expression[33], potentially explaining the lack of change in circadian activity in our study. In support of this hypothesis, dysregulated CORT infusion pattern did not affect any *clock* genes in the suprachiasmatic nucleus. In addition, the gene expression of *Hcrt* in the lateral hypothalamic area, which plays an important physiological role in regulating the sleep-wake state[54] was not affected by dysregulated CORT infusion pattern. Taken together, these data indicate that suprachiasmatic nucleus mediated circadian activity is primarily regulated by light:dark cues, and is unaffected by CORT.

Intriguingly, the *out-of-phase* group experienced a significant elevation in CBT at a time coinciding with the high amplitude CORT pulses delivered at the start of the inactive phase. The elevated temperature could not be simply assigned to post-prandial hyperthermia, as we also found a similar temperature rise in fasting rats treated with an acute CORT treatment (S Fig. 6). While the precise mechanism of elevated temperature is yet to be resolved, it most likely involves a combination of central regulation of thermogenesis and downstream peripheral metabolic effects[55–59]. Nevertheless, this represents an important finding, especially as a similar misalignment between daily activity cycles, body temperature and cortisol has recently been reported in nurses who work night shifts[60].

In conclusion, a normal circadian profile of CORT secretion is essential for appropriate timing of food intake. This has significant implications for human physiology in situations of shift work, jet lag, and chronic sleep disturbance; all of which have been found to be associated with adverse metabolic effects[61–63]. Our data contribute to an improved understanding of the mechanisms underlying these metabolic effects and should help provide a framework for a rational approach for prevention of these disorders and for developing improved therapeutics.

## Materials and methods

**Animals and ethical approval**. Adult male *Sprague-Dawley (SD)* rats (220–240 g, Harlan-Olac, Oxon, UK) were singly housed with ad libitum access to food and water or 0.9% physiological saline water. They were kept under 12:12 light/dark (light on at 08.00 h) with normal laboratory conditions. We have complied with all relevant ethical regulations for animal testing. All animal procedures were carried out in accordance with the ARRIVE guidelines[64,65], UK Home Office animal welfare regulations, and approved by written consent from the Animal Welfare and Ethical Review Body (University of Bristol).

**Bilateral adrenalectomy, and implantation of intravenous cannulae and telemetry probes**. Intravenous cannulation of the jugular vein was performed as described[8]. Briefly, after deep anaesthesia with isofluorane the right jugular vein was cannulated by inserting a polythene cannulae (Scientific Commodities, Lake Havasu City, Arizon, US). During the same surgery, bilateral adrenal glands were removed by the dorsal approach. A telemetry probe (Nanotag®, Kissei Comtec, Japan) that recorded both locomotor activity and core body temperature at 5 min intervals was implanted into the peritoneal cavity. Nonsteroidal anti-

inflammatory drug (5 mg/kg of rimadyl®, Zoetis, U.S.A.) and glucose saline (100 ml/kg) were administered peri-operatively by subcutaneous injection. Animals were individually housed in specially designed twin-sensory cages (Tecniplast UK) with counter balanced tethers attaching the infusion cannulae to 360° mechanical swivels that allowed maximal freedom of movement. Animals had free access to food and 0.9% saline throughout the experiment. Sham-operated control rats were subjected to the same incisions as the adrenalectomized rats, but the adrenal glands were not removed.

**CORT-HBC delivery by programmable infusion pump.** A schematic pattern of the CORT-HBC (Sigma-Aldrich, U.S.A.) delivery can be found in Fig. 1. CORT-HBC was dissolved in 0.9% heparinized saline (10 U/mL heparin; CP Pharmaceuticals Ltd, UK). One day after surgery, at ZT12, animals were connected to an automated infusion system (Harvard syringe pump®, Harvard Apparatus, U.S.A.). The dose of CORT delivered was chosen to match average endogenous circulating CORT determined from our automated blood sampling system of adrenal-intact rats (S Fig. 1). Animals ($n = 8$ in each group) were infused with either *in-phase* or *out-of-phase* CORT (Fig. 1). Both groups were infused with the same total amount of CORT (area under the curve, 0.92 mg CORT over 24 h). Frequency of one portion of pulsatile infusion was 60 min with 20 min on-rate and 40 min off-rate.

**Measurement of food intake.** All rats were handled for 5 days before the experiment. They were housed in twin sensory cages 2 days before the surgery to habituate. One day following the surgery, CORT infusions started at ZT12, and measurement of food intake commenced. Food intake was measured manually every 12 h for 5 days.

**Sample collection.** After 5 days of CORT-HBC pulsatile infusion, rats were decapitated under terminal anaesthesia (inhalable isofluorane) and trunk blood, brains, and pituitaries were collected at ZT1 and ZT13 for further analysis. Blood samples were collected in 50 mL conical-bottom centrifuge tube containing ethylene diamine tetra acetic acid (0.5 M, pH 7.4) on ice. Brains and pituitaries were frozen immediately on dry ice.

**Measurement of ACTH and CORT.** Blood samples were centrifuged for 15 min at 4000 rpm at 4 °C. Plasma was stored at −70 °C until assayed. Plasma ACTH levels were determined, using 100 μL, via immunoradiometric assay (Diasorin Ltd., Dartford, UK) according to the manufacturer's instructions. Total CORT was measured in the trunk blood plasma samples using our well established in house radio-immuno assay[8,66]. Antiserum was kindly donated by G. Makara (Institute of Experimental Medicine, Budapest, Hungary). [125I]-corticosterone was obtained from Izotop (Institute of Isotopes Co. Ltd., Budapest, Hungary) with a specific activity of 2 mCi/mL. The sensitivity of the assay at the blood dilutions used was 1 ng/mL. All samples were processed in the same assay to exclude inter-assay variability. Intra-assay coefficients of variation are 2.8 and 7.3% for the ACTH and CORT assay, respectively[9].

**In situ hybridization histochemistry.** Brains were cut into coronal 12 μm sections, and thaw mounted on gelatin/chrome alum-coated slides. The locations of the nuclei including the suprachiasmatic nucleus, supraoptic nucleus, paraventricular nucleus, arcuate nucleus, lateral hypothalamic area were determined according to the coordinates in the rat brain atlas[67]. 35S 3′-end-labelled deoxyoligonucleotide complementary probe sequences to

transcripts encoding *Per1, Per2, hnAvp, Avp, Oxt, Crh, Pomc, Cartpt, Npy, Agrp, Mch* and *Hcrt* were used.

*Per1*:CTCTTGTCAGGAGGAATCCGGGGGAGCTTCATAAC-CAGAGTGGATG

*Per2*:GTGGCCTTCCGGGATGGGATGTTGGCTGG-GAACTCGCACTTTCTT

*hnAvp*:GCACTGTCAGCAGCCCTGAACGGACCA-CAGTGGTAC

*Avp*:CAGCTCCCGGGCTGGCCCGTCCAGCT

*Oxt*:CTCGGAGAAGGCAGACTCAGGGTCGCAGGC

*Crh*:CAGTTTCCTGTTGCTGTGAGCTTGCTGAGC-TAACTGCTCTGCCCTGGC

*Pomc*:CTTCTTGCCCACCGGCTTGCCCCAGCA-GAAGTGCTCCATGGACTAGGA

*Cartpt*:TGGGGACTTGGCCGTACTTCTTCTCATAGATCG-GAATGCG

*Npy*:GGAGTAGTATCTGGC-CATGTCCTCTGCTGGCGCGTC

*Agrp*:CGACGCGGAGAACGAGACTCGCGGTTCTGTG-GATCTAGCACCTCTGCC

*Pmch*:CCAACAGGGTCGGTAGACTCGTCCCAGCAT

*Hcrt*:TTCGTAGAGACGGCAGGAACACGTCTTCTGGC-GACA

Our well established ISHH protocol was used[41,68,69]. Briefly sections were fixed in 4% (w/v) formaldehyde for 5 min and incubated in saline containing 0.25% (v/v) acetic anhydride and 0.1 M triethanolamine for 10 min. Sections were then dehydrated in ethanol, delipidated in chloroform, and partially rehydrated. Hybridization was carried out overnight at 37 °C in 45 μL of hybridization buffer under a Nescofilm (Bando Kagaku, Osaka, Japan) cover slip. A total count of $1 \times 10^5$ c.p.m. for Oxt and Avp transcripts, $1 \times 10^6$ c.p.m. for *Per1, Per2, hnAvp, Crh, Pomc, Cartpt, Npy, Agrp, Pmch* and *Hcrt* transcripts per slide were used for ISHH. After hybridization, sections were washed 4 times with SSC (150 mM NaCl and 15 mM sodium citrate) for 1 hour at 65 °C and for an additional hour with two changes of SSC at room temperature. Hybridized sections were exposed for autoradiography (Hyperfilm, Amersham, Bucks, UK) for 6 hours for *Oxt* probe, 12 hours for *Avp* probe, 5 days for *Pmch* and *Hcrt* probes, and 1 week for *Per1, Per2, hnAvp, Crh, Pomc, Cartpt, Npy* and *Agrp* probes. The amount of bound probe to mRNA was analysed in comparison to $^{14}$C-labelled standards (Amersham, Bucks, UK) using image analysis software (NIH Image 1.6.2, W.Rasband, NIH, Bethesda, MD, USA). The obtained results were represented in arbitrary units setting the mean optical density (OD) obtained from sham-operated control rats. Representative images are shown in our supplementary information (S Figs. 4, 5, and 7).

**Chromatin immunoprecipitation assays: tissue collection.** Adrenalectomized rats were injected with CORT-HBC (3 mg/kg; subcutaneous) at ZT1. After exactly 1 h post-injection, rats were decapitated under terminal anaesthesia (inhalable isofluorane) and brains dissected using a coronal brain matrix (Harvard apparatus) to collect $3\text{mm}^2$ brain sections containing hypothalamic arcuate nucleus.

The tissue was immediately fixed according to our well established protocol[8,70] in 1% (v/v) formaldehyde (Sigma, UK), phosphate buffered saline (1.37 M NaCl, 2.68 mM KCl, 10.14 mM Na$_2$HPO$_4$, pH 7.4) solution for 10 min at room temperature. Formaldehyde cross-linking was quenched with addition of glycine (final conc. 125 mM) for 5 min and washed three times in ice cold phosphate buffered saline supplemented with 2 mM NaF, 0.2 mM sodium orthovanadate and 1X cOmplete protease inhibitor (Roche Diagnostics). Fixed tissue

was stored at −70 °C in 500 μl of S1 Buffer (10 mM HEPES, pH 7.9, 10 mM KCl, 15 mM MgCl$_2$, 0.1 mM (EDTA), pH 8) supplemented with 0.5 mM Dithiothreitol and 2 mM NaF, 0.2 mM sodium orthovanadate and 1X cOmplete protease inhibitor.

**Chromatin immunoprecipitation assays: Chromatin fragmentation.** Samples were thawed slowly on ice and adjusted to a final volume of 1 ml with supplemented S1 buffer and Dounce homogenized. Lysate was centrifuged at 2000xg (4 °C) and lysed in supplemented (2 mM NaF and 0.2 mM sodium orthovanadate and 1X cOmplete protease inhibitor) sodium dodecyl sulphate (SDS) lysis buffer (2% SDS, 10 mM EDTA, 50 mM Tris-HCl (pH 8.1)). Soluble chromatin was prepared according to a protocol developed in house[71,72] using MNase digestion [50 mM Tris-HCl, pH 7.5, 4 mM MgCl$_2$, 1 mM CaCl$_2$, 0.32 mM sucrose, 2 mM NaF, 0.2 mM NaVan] with 2 units of MNase (Sigma, UK)/180 μg chromatin.

**Chromatin immunoprecipitation assay.** ChIP buffers were prepared in house, as described in the EZ ChIP kit protocol (Upstate Bio- technology, Lake Placid, NY, USA) with some modifications for use with brain tissue. MNase digested chromatin was diluted to 50 μg in 100 μl of SDS lysis buffer and 0.9 ml with supplemented (2 mM NaF, 0.2 mM NaVan and 1X cOmplete protease inhibitor) ChIP dilution buffer (0.01% SDS, 1.1% Triton X-100, 1.2 mM EDTA, 16.7 mM Tris-HCl pH 8.1, 167 mM NaCl). After a pre-clearing step with protein A Dynabeads to reduce non-specific binding, samples were immunoprecipitated with either anti-GR antibody [2 μg; catalogue number 24050-1-AP (Proteintech)], anti-pSer5 RNA Pol2 [10 μl; catalogue number 39233 (ACTIVE MOTIF)] or normal rabbit IgG [2 μg; catalogue number 2729 S (Cell Signalling Technology)], incubated overnight at 4 °C. DNA-Protein-antibody complexes were captured by incubation with protein A Dynabeads for 4 h at 4 °C (Sigma-Aldrich, UK), followed by sequentially washes with 150 mM salt buffer (0.1% SDS, 1% Triton X-100, 2 mM EDTA, 20 mM Tris-HCl pH 8.1), 500 mM salt buffer (0.1% SDS, 1% Triton X-100, 2 mM EDTA, 20 mM Tris-HCl pH 8.1), LiCl buffer (0.25 M LiCl, 1% IGEPAL-CA630, 1% deoxycholic acid sodium salt, 1 mM EDTA, 10 mM Tris-HCl pH 8.1) and TE buffer (10 mM Tris-HCl, 1 mM EDTA pH 8.0). Complexes were eluted from the Dynabeads in 1% SDS 100 mM and 0.1 M NaHCO3. NaCl was added (300 mM final concentration) and crosslinks reversed overnight at 65 °C. RNA was removed using RNase treatment (Roche Diagnostics) at 65 °C and protein was digested using proteinase K (Ambion, Huntington, UK) after adjusting each solution with EDTA (1 mM final) and Tris-HCl (4 mM final) at 45 °C. DNA was extracted using 25:24:1 phenol-chloroform-isoamyl alcohol (Sigma, UK) followed by 24:1 chloroform-isoamyl alcohol (Sigma, UK). DNA in the aqueous phase was precipitated overnight at -20 °C in 2.5 VOL 100% ethanol and 20 μg glycogen (Sigma-Aldrich, UK). DNA was pelleted by centrifugation at 13,000 rpm, 4 °C and washed in 70% Ethanol (13,000 rpm, 4 °C), air dried and suspended in 40 μl nuclease free water (Ambion, Huntington, UK).

**Quantitative real time PCR.** RT-qPCR was performed using PowerUp SYBR Green Master Mix (A25742; Applied Biosystems) and the following primers:

rPer1_GRE_For: CCAAGGCTGAGTGCATGTC
rPer1_GRE_Rev: GCGGCCAGCGCACTA
rPer1_TSS_For: TGGCTGATGACACTGATGCAA
rPer1_TSS_Rev: GAGCTGAGTCCTTGCCATTG

rNpy_GRE_For: TAGACCGCATGTGGAGAACC
rNpy_GRE_Rev: TGGGTTTGAGTGAAAGGGGG
rNpy_TSS_For: CGCTCCATAAAAGCCCGTTG
rNpy_TSS_Rev: CTGCGAGGAATGAGCTCCAC

**Statistics and reproducibility.** All data shown are from individual rats (biological replicates; number of replicates are indicated in the figure legend for each experiment). Rats were randomly assigned in all experiments. The results were double-blinded in all experiments. All data were tested for normal distribution using D'Agnostino & Pearson, Anderson-Darling, Shapiro-Wilk and Kolmogorov-Smirnov tests, and found to pass normality test (alpha = 0.05) in at least one test. Data were then analysed by 2-WAY ANOVA, either with repeated measures for cumulative food and water intake, locomotor activity and body temperature, or ordinary 2-WAY ANOVA with time and treatment as the main factors for the ISHH data analyses. When a main effect of treatment or time, or an interaction between treatment and time was found by 2-WAY ANOVA, Bonferroni multiple comparison post-tests were performed to identify significant one-to-one differences between timepoints (within the same treatment) shown on the graphs as $*p < 0.05$, $**p < 0.01$, $***p < 0.001$,$****p < 0.0001$ and significant differences between treatment (at the same timepoint) shown on the graphs as $^\$p < 0.05$, $^{\$\$}p < 0.01$, $^{\$\$\$}p < 0.001$,$^{\$\$\$\$}p < 0.0001$. For the ChIP data, which measured CORT-induced effects relative to saline-treated controls, unpaired $T$ Test with Welch's correction for unequal variances was used, and significant differences were shown on the graphs $*p < 0.05$, $**p < 0.01$, $***p < 0.001$,$****p < 0.0001$. In all cases, mean ± SEM was plotted graphically.

**Reporting summary.** Further information on research design is available in the Nature Portfolio Reporting Summary linked to this article.

### Data availability

All data supporting the findings of this study are available within the paper and in Supplementary Information (Supplementary Data 1). Further experimental data and materials are available from the corresponding author (or other sources, as applicable) upon reasonable request.

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

## Acknowledgements

This paper was supported by a Grant-in-Aid for Scientific Research (C) (21K06779) KAKENHI for M.Y. from the Ministry of Education, Culture, Sports, Science, and Technology (MEXT), Japan; and a University of Occupational and Environmental Health (UOEH) Grant-in-Aid for Priority Research in the Field of Occupational Medicine (2021, 2022) for M.Y. from the UOEH, Japan. S.L., B.CC., B.F., Y.K., Z.Z. were supported by United Kingdom Medical Research Council grant MR/R010919/1. Costs of all animal work and experimental reagents and consumables were supported by United Kingdom Medical Research Council grant MR/R010919/1.

## Author contributions

This study was designed by M.Y[1,2], B.F.[1], B.CC.[1]. Experiments were performed by M.Y.[1,2], B.F.[1], Y.K.[1], Z.Z.[1], and B.CC.[1]. Data were analysed and interpreted by M.Y.[1,2], B.CC.[1], Y.U.[2], and S.L.[1]. Draft and Figures are prepared by M.Y.[1,2] and B.CC[1]. Final Approval was made by S.L.[1]. All authors approved the final version of the manuscript and agreed to be accountable for all aspects of the work in ensuring that questions related to the accuracy. All authors designated as authors qualify for authorship, and all those who qualify for authorship are listed.

## Competing interests

The authors declare no competing interests.

## Additional information

Copyright PermissionLicense Agreement between Dr Becky L. Conway-Campbell / University of Bristol and Copyright Clearance Centre, Inc. ("CCC") on behalf of Elsevier Science & Technology Journals for use of image/photo/illustrations from 'The rat brain in stereotaxic coordinates'[67] (Fourth edition, published 01/01/1998). License ID 1394840-1. ISBN-13. 9780125476195.

