## [Peer Review File · Communications Biology]

Reviewers' comments:

Reviewer #1 (Remarks to the Author):

The article entitled "Phase-shifting the circadian glucocorticoid profile induces disordered feeding behavior by dysregulating hypothalamic neuropeptide gene expression" by Yoshimura and colleagues describes feeding behavior in adrenalectomized rats infused with corticosterone in- and out-of-phase with circadian lighting cues. Results are clear and novel. Expression of neuropeptides regulating food intake nicely fits the feeding phenotype. The study is well conducted and data well presented. However, a direct CORT effect on the expression of neuropeptides expressed by different hypothalamic nuclei is not fully demonstrated.

Major comments:

- 1) Fig 2B: please, indicate the day after surgery in which the 12h food intake was measured.
- 2) Fig 2B and D: please, clarify in the figure legend which statistical test was used for one-to-one comparison.
- 3) Fig 2F: The quality of the pictures showing the overview of the brain region analyzed is extremely low and needs to be improved. In the current version is hard to see ARC or LHA. Additionally a negative control picture is necessary here and a description of the size of the scale bar in the figure legend. Also applies for Fig 3B and 4E
- 4) Fig 3C: Figure legend mentions a comparison (\$\$) between (one can assume) out-of-phase ZT13 vs ZT13 in-phase? The comparison is missing in the figure. Please, correct it.
- 5) Fig 4A and 4C: Units for locomotor activity have to be indicated
- 6) Statistics: Tests used to probe normality before ANOVA analysis should be mentioned and information about the test results have to be included at least in the figure legend to assess the size of the differences observed.
- 7) Figure legends contain a description of the results, which might be redundant with the result section. As mentioned above, would be more informative to have in the legend the results of the statistical tests.
- 8) Page 10 line 228: Authors referred to a change in appetitive behavior during the inactive phase, however only food intake was measured every 12 hs and the correlation with the change in CORT peak might be not happening at the same time. Please, revise this statement.
- 9) Authors suggested a role of GR however, is it possible to exclude MR-dependent effects?
- 10) Page 14 line 328-330: this sentence needs references
- 11) Page 14 line 332-333: Authors need to explain what they mean with "rational approach improved therapeutics"
- 12) More elaborated discussion on the changes observed in core body temperature with no changes in activity is necessary.

13) Levels of peripheral hormones regulating food intake might be necessary to really explain whether the effects on food intake are directly or indirectly caused by CORT (page 12 lines 276-278). If possible, collecting these data should be considered.

14) In order to attribute to CORT the changes in feeding rhythms and metabolic phenotype authors would need to investigate food intake at a later time point after surgery. Could be possible that SCN driven rhythms become more dominant later on and restore the feeding rhythms?. If possible, collecting these data should be considered.

15) Please, add proper reference to supplementary figures and supplementary figures legends.

Minor changes

16) Control group has to be called either control or sham in a consistent way throughout the paper

17) Page 4 line 76: POMC and CART abbreviations need to be explained

18) Page 4 line 79 and 82: NPY abbreviation was already explained above

19) Page 4 line 86: should read "current study"

20) Page 4 line 87: should read "we investigate.....associated with phase shifting circadian glucocorticoid profile" since there is no shift work or transmeridian travel effects assessed.

21) Page 5 line 114: "highly disordered" food intake sounds inappropriate to describe a 12-h time resolution in food intake.

22) Page 7 lines 164 and 165: Consistent is written twice in two consecutive sentences.

23) Color coding could be confusing. As a suggestion, out-of-phase group could be either red or black and red could be avoided for the IHH results might help.

24) Page 17 line 391: cort should read CORT

Reviewer #2 (Remarks to the Author):

This manuscript by Mitsuhiro Yoshimura et al. examined the effects of Phase-shifting the circadian glucocorticoid profile on energy metabolism and feeding behavior in rats to reveal the relationship between circadian glucocorticoid and hypothalamic neuropeptide gene in the process of eating disorder. Besides, the authors have shown that appropriate timing of food intake may be associated with the observed normal circadian profile of CORT secretion.

Overall this is a straightforward study on a fascinating topic. Additionally, there are several advantages of circadian on energy metabolism shown in the last several years, and this one is a continuation of one of such studies. I have the following reservations with this study, and they had to be addressed.

1. It seems like these authors have selectively picked up few genes without providing too much logistics and background. The logistics checking anorexigenic and orexigenic neuropeptide gene expression level is not defined appropriately.

2. The authors have also jumped the here and there without providing any mechanistic detail. For

example, they are increasing NPY, AgRP, POMC, and CART expression in brain. What is the relationship of CORT with these genes?

3. Additionally, it will be essential to demonstrate cytological evidence to reveal the impact of CORT disorder, for example, mitochondrial function/integrity, oxidative stress. This might help to demonstrate the mechanistic effect of disordered feeding behaviour for the obesity models.

4. Several places author has indicated that Phase-shifting the circadian glucocorticoid profile induces food intake, yet the mechanisms underlying its detailed effects are not fully understood. I have failed to see any mechanism in this paper. The authors in tone in the entire manuscript about these kinds of statements.

Reviewer #1 (Remarks to the Author):

The article entitled "Phase-shifting the circadian glucocorticoid profile induces disordered feeding behavior by dysregulating hypothalamic neuropeptide gene expression" by Yoshimura and colleagues describes feeding behavior in adrenalectomized rats infused with corticosterone in- and out-of-phase with circadian lighting cues. Results are clear and novel. Expression of neuropeptides regulating food intake nicely fits the feeding phenotype. The study is well conducted and data well presented. However, a direct CORT effect on the expression of neuropeptides expressed by different hypothalamic nuclei is not fully demonstrated.

Major comments:

1) Fig 2B: please, indicate the day after surgery in which the 12h food intake was measured.

Thank you for your indication. Surgery was carried out on Day 0. Corticosterone infusions commenced on Day 1. Twelve-hourly food intake was measured until day 4. Recording of food intake commenced on Day 1 of Infusion, therefore the effects of infusion on feeding behavior was immediate. There was no significant difference in food intake in a comparison between Day 1, Day 2, Day 3 and Day 4, indicating that feeding behavior did not change over time, at least not for the duration of the experiment. Now, individual data plot was added for Fig. 2. Please see in lines 120-121, on page 6. Please also see Fig. 2 and its legend on page 35.

2) Fig 2B and D: please, clarify in the figure legend which statistical test was used for one-to-one comparison.

The figure legend has been updated. Statistical test used in each figure is now described in better detail in "Statistical analysis". Please see in lines 609-625, on pages 26 and 27.

3) Fig 2F: The quality of the pictures showing the overview of the brain region analyzed is extremely low and needs to be improved. In the current version is hard to see ARC or LHA. Additionally a negative control picture is necessary here and a description of the size of the scale bar in the figure legend. Also applies for Fig 3B and 4E

Thank you very much for your indication. We included the ISH overview images to show the overall morphology of the imaged region. Higher resolution images of the low magnification overviews are not available, so we have included (in Figure 2e) an image from the rat brain atlas indicating the location of the ARC and LHA to help the readers understand the neuroanatomical features of interest. Regarding your query about negative control pictures for ISH, unfortunately we are unable to provide these. Although we realize this does not answer your indication, the authors are experienced with the ISH protocol, and all antisense probes which were used in the present study have been confirmed by many previous studies that are already published (See References below).

References (rationale for using confirmed probes for feeding regulating peptide mRNA in ISHH)

1. Tanaka K, Saito R, Sanada K, Nishimura H, Nishimura K, Sonoda S, Ueno H, Motojima Y, Matsuura T, **Yoshimura M**, Maruyama T, Onaka T, Yamamoto Y, Kusuhara K, **Ueta Y**. Expression of hypothalamic feeding-related peptide genes and neuroendocrine responses in an experimental allergic encephalomyelitis rat model. *Peptides*. 2020 Jul;129:170313. doi: 10.1016/j.peptides.2020.170313. PMID: 32298774 (2020)
2. Sonoda S, **Yoshimura M**, Ueno H, Nishimura H, Nishimura K, Tanaka K, Motojima Y, Saito R, Maruyama T, Hashimoto H, Okada Y, Tanaka Y, **Ueta Y**. Expression of the genes encoding hypothalamic feeding-related neuropeptides in the streptozotocin-induced diabetic rats with variable hyperglycemia and hyperphagia. *Neuropeptides*. 2019 Mar 20. pii: S0143-4179(18)30185-9. doi: 10.1016/j.npep.2019.03.003. PMID: 30928158 (2019)
3. Sonoda S*, **Yoshimura M***, Abe C, Morita H, Ueno H, Motojima Y, Saito R, Maruyama T, Hashimoto H, Tanaka Y, **Ueta Y**. Effects of hypergravity on the gene expression of the hypothalamic feeding-related neuropeptides in mice via vestibular inputs. *Peptides*. 2018 May 8;105:14-20. doi: 10.1016/j.peptides.2018.05.004. PMID: 29751050 *equally contributed as first authors. (2018)
4. Arase K, Hashimoto H, Sonoda S, Ueno H, Saito R, Motojima Y, **Yoshimura M**, Maruyama T, Hirata K, Uezono Y, **Ueta Y**. Possible involvement of central oxytocin in cisplatin-induced anorexia in rats. *J Physiol Sci*. 2017 Jun 14. doi: 10.1007/s12576-017-0550-z. PMID: 28616820 (2017)
5. Terawaki K, Kashiwase Y, Sawada Y, Hashimoto H, **Yoshimura M**, Ohbuchi K, Sudo Y, Suzuki M, Miyano K, Shiraishi S, Higami Y, Yanagihara K, Hattori T, Kase Y, **Ueta Y**, Uezono Y. Development of ghrelin resistance in a cancer cachexia rat model using human gastric cancer-derived 85As2 cells and the palliative effects of the Kampo medicine rikkunshito on the model. *PLoS One*. 2017 Mar 1;12(3):e0173113. doi: 10.1371/journal.pone.0173113. PMID: 28249026 (2017)
6. So M, Hashimoto H, Saito R, Yamamoto Y, Motojima Y, Ueno H, Sonoda S, **Yoshimura M**, Maruyama T, Kusuhara K, **Ueta Y**. Inhibition of ghrelin-induced feeding in rats by pretreatment with a novel dual orexin receptor antagonist. *J Physiol Sci*. 2017 Jan 4. doi: 10.1007/s12576-016-0517-5. PMID: 28054308 (2017)
7. Terawaki K, Sawada Y, Kashiwase Y, Hashimoto H, **Yoshimura M**, Suzuki M, Miyano K, Sudo Y, Shiraishi S, Higami Y, Yanagihara K, Kase Y, **Ueta Y**, Uezono Y. New cancer cachexia rat model generated by implantation of a peritoneal dissemination-derived human stomach cancer cell line. *Am J Physiol Endocrinol Metab*. 2014 Feb 15;306(4):E373-87. doi: 10.1152/ajpendo.00116.2013. PMID: 24347053 (2014)
8. **Yoshimura M**, Hagimoto M, Matsuura T, Ohkubo J, Ohno M, Maruyama T, Ishikura T, Hashimoto H, Kakuma T, Yoshimatsu H, Terawaki K, Uezono Y, Toyohira Y, Yanagihara N, **Ueta Y**. Effects of food deprivation on the hypothalamic feeding-regulating peptides gene expressions in serotonin depleted rats. *J Physiol Sci*. 2014 Mar;64(2):97-104. doi: 10.1007/s12576-013-0296-1. PMID: 24162946

(2014)

9. **Yoshimura M**, Matsuura T, Ohkubo J, Ohno M, Maruyama T, Ishikura T, Hashimoto H, Kakuma T, Yoshimatsu H, Terawaki K, Uezono Y, **Ueta Y**. The gene expression of the hypothalamic feeding-regulating peptides in cisplatin-induced anorexic rats. *Peptides*. 2013 Aug;46:13-9. doi: 10.1016/j.peptides.2013.04.019. PMID: 23684922 (2013)

References (where RI ISHH was used)

1. Birnie MT, Claydon MDB, Troy O, Flynn BP, **Yoshimura M**, Kershaw YM, Zhao Z, Demski-Allen RCR, Barker GRI, Warburton EC, Bortolotto ZA, **Lightman SL**, **Conway-Campbell BL**. Circadian regulation of hippocampal function is disrupted with corticosteroid treatment. *Proc Natl Acad Sci U S A*. 2023 Apr 11;120(15):e2211996120. doi: 10.1073/pnas.2211996120. Epub 2023 Apr 6. PMID: 37023133
2. Nishimura H*, **Yoshimura M***, Shimizu M, Sanada K, Sonoda S, Nishimura K, Baba K, Ikeda N, Motojima Y, Maruyama T, Nonaka Y, Baba R, Onaka T, Horishita T, Morimoto H, Yoshida Y, Kawasaki M, Sakai A, Muratani M, **Conway-Campbell B**, **Lightman S**, **Ueta Y**#. Endogenous oxytocin exerts anti-nociceptive and anti-inflammatory effects in rats. *Commun Biol*. 2022 Sep 5;5(1):907. doi: 10.1038/s42003-022-03879-8. PMID: 36064593 *equally contributed as first authors. #corresponding authors. (2022)
3. Sanada K, Ueno H, Miyamoto T, Baba K, Tanaka K, Nishimura H, Nishimura K, Sonoda S, **Yoshimura M**, Maruyama T, Onaka T, Otsuji Y, Kataoka M, **Ueta Y**. AVP-eGFP was significantly upregulated by hypovolemia in the parvocellular division of the paraventricular nucleus in the transgenic rats. *Am J Physiol Regul Integr Comp Physiol*. 2022 Jan 12. doi: 10.1152/ajpregu.00107.2021. PMID: 35018823 (2022)
4. Ueno H, Sanada K, Miyamoto T, Baba K, Tanaka K, Nishimura H, Nishimura K, Sonoda S, **Yoshimura M**, Maruyama T, Oginosawa Y, Araki M, Sonoda S, Onaka T, Otsuji Y, **Ueta Y**. Oxytocin-monomeric red fluorescent protein 1 synthesis in the hypothalamus under osmotic challenge and acute hypovolemia in a transgenic rat line. *Physiol Rep*. 2020 Sep;8(17):e14558. doi: 10.14814/phy2.14558. PMID: 32914562 (2020)
5. Nishimura H, Kawasaki M, Suzuki H, Matsuura T, Baba K, Motojima Y, Yamanaka Y, Fujitani T, Ohnishi H, Tsukamoto M, Maruyama T, **Yoshimura M**, Nishimura K, Sonoda S, Sanada K, Tanaka K, Onaka T, **Ueta Y**, Sakai A. The neurohypophysial oxytocin and arginine vasopressin system is activated in a knee osteoarthritis rat model. *J Neuroendocrinol*. 2020 Aug;32(8):e12892. doi: 10.1111/jne.12892. PMID: 32761684 (2020)
6. Akiyama Y, **Yoshimura M**, Ueno H, Sanada K, Tanaka K, Sonoda S, Nishimura H, Nishimura K, Motojima Y, Saito R, Maruyama T, Hirata K, Uezono Y, **Ueta Y**. Peripherally administered cisplatin activates a parvocellular neuronal subtype expressing arginine vasopressin and enhanced green fluorescent protein in the paraventricular nucleus of a transgenic rat. *J Physiol Sci*. 2020 Jul 10;70(1):35. doi: 10.1186/s12576-020-00764-z. PMID: 32650712 (2020)
7. Tanaka K, Saito R, Sanada K, Nishimura H, Nishimura K, Sonoda S, Ueno H, Motojima Y, Matsuura T, **Yoshimura M**, Maruyama T, Onaka T, Yamamoto Y, Kusuhara K, **Ueta Y**. Expression of hypothalamic feeding-related peptide genes and neuroendocrine responses in an experimental allergic encephalomyelitis rat model. *Peptides*. 2020 Jul;129:170313. doi: 10.1016/j.peptides.2020.170313. PMID: 32298774 (2020)
8. Nishimura H, Kawasaki M, Matsuura T, Suzuki H, Motojima Y, Baba K, Ohnishi H, Yamanaka Y, Fujitani T, **Yoshimura M**, Maruyama T, Ueno H, Sonoda S, Nishimura K, Tanaka K, Sanada K, Onaka T, **Ueta Y**, Sakai A. Acute Mono-Arthritis Activates the Neurohypophysial System and Hypothalamo-Pituitary Adrenal Axis in Rats. *Front Endocrinol (Lausanne)*. 2020 Feb 11;11:43. doi: 10.3389/fendo.2020.00043. PMID: 32117068 (2020)
9. Ueno H, Serino R, Sanada K, Akiyama Y, Tanaka K, Nishimura H, Nishimura K, Sonoda S, Motojima Y, Saito R, **Yoshimura M**, Maruyama T, Miyamoto T, Tamura M, Otsuji Y, **Ueta Y**. Effects of acute kidney dysfunction on hypothalamic arginine vasopressin synthesis in transgenic rats. *J Physiol Sci*. 2019 May;69(3):531-541. doi: 10.1007/s12576-019-00675-8. PMID: 30937882 (2019)

10. Sonoda S, **Yoshimura M**, Ueno H, Nishimura H, Nishimura K, Tanaka K, Motojima Y, Saito R, Maruyama T, Hashimoto H, Okada Y, Tanaka Y, **Ueta Y**. Expression of the genes encoding hypothalamic feeding-related neuropeptides in the streptozotocin-induced diabetic rats with variable hyperglycemia and hyperphagia. *Neuropeptides*. 2019 Mar 20. pii: S0143-4179(18)30185-9. doi: 10.1016/j.npep.2019.03.003. PMID: 30928158 (2019)
11. Nishimura K, Yoshino K, Sanada K, Beppu H, Akiyama Y, Nishimura H, Tanaka K, Sonoda S, Ueno H, **Yoshimura M**, Maruyama T, Ozawa H, **Ueta Y**. Effect of oestrogen-dependent vasopressin on HPA axis in the median eminence of female rats. *Sci Rep*. 2019 Mar 26;9(1):5153. doi: 10.1038/s41598-019-41714-z. PMID: 30914732 (2019)
12. Nishimura H, Kawasaki M, Suzuki H, Matsuura T, Motojima Y, Ohnishi H, Yamanaka Y, **Yoshimura M**, Maruyama T, Saito R, Ueno H, Sonoda S, Nishimura K, Onaka T, **Ueta Y**, Sakai A. Neuropathic Pain Up-Regulates Hypothalamo-Neurohypophysial and Hypothalamo-Spinal Oxytocinergic Pathways in Oxytocin-Monomeric Red Fluorescent Protein 1 Transgenic Rat. *Neuroscience*. 2019 Mar 1;406:50-61. doi: 10.1016/j.neuroscience.2019.02.027. PMID: 30826522 (2019)
13. Sonoda S*, **Yoshimura M***, Abe C, Morita H, Ueno H, Motojima Y, Saito R, Maruyama T, Hashimoto H, Tanaka Y, **Ueta Y**. Effects of hypergravity on the gene expression of the hypothalamic feeding-related neuropeptides in mice via vestibular inputs. *Peptides*. 2018 May 8;105:14-20. doi: 10.1016/j.peptides.2018.05.004. PMID: 29751050 *equally contributed as first authors. (2018)
14. Ueno H, **Yoshimura M**, Tanaka K, Nishimura H, Nishimura K, Sonoda S, Motojima Y, Saito R, Maruyama T, Miyamoto T, Serino R, Tamura M, Onaka T, Otsuji Y, **Ueta Y**. Upregulation of hypothalamic arginine vasopressin by peripherally administered furosemide in transgenic rats expressing arginine vasopressin-enhanced green fluorescent protein. *J Neuroendocrinol*. 2018 Apr 22:e12603. doi: 10.1111/jne.12603. PMID: 29682811 (2018)
15. Ohno S, Hashimoto H, Fujihara H, Fujiki N, **Yoshimura M**, Maruyama T, Motojima Y, Saito R, Ueno H, Sonoda S, Ohno M, Umezumi Y, Hamamura A, Saeki S, **Ueta Y**. Increased oxytocin-monomeric red fluorescent protein 1 fluorescent intensity with urocortin-like immunoreactivity in the hypothalamo-neurohypophysial system of aged transgenic rats. *Neurosci Res*. 2017 Aug 30. pii: S0168-0102(17)30164-5. doi: 10.1016/j.neures.2017.08.001. PMID: 28859972 (2017)
16. Arase K, Hashimoto H, Sonoda S, Ueno H, Saito R, Motojima Y, **Yoshimura M**, Maruyama T, Hirata K, Uezono Y, **Ueta Y**. Possible involvement of central oxytocin in cisplatin-induced anorexia in rats. *J Physiol Sci*. 2017 Jun 14. doi: 10.1007/s12576-017-0550-z. PMID: 28616820 (2017)
17. Motojima Y, Matsuura T, **Yoshimura M**, Hashimoto H, Saito R, Ueno H, Maruyama T, Sonoda S, Suzuki H, Kawasaki M, Ohnishi H, Sakai A, **Ueta Y**. Comparison of the induction of c-fos-eGFP and Fos protein in the rat spinal cord and hypothalamus resulting from subcutaneous capsaicin or formalin injection. *Neuroscience*. 2017 Jul 25;356:64-77. doi: 10.1016/j.neuroscience.2017.05.015. PMID: 28527956 (2017)
18. Terawaki K, Kashiwase Y, Sawada Y, Hashimoto H, **Yoshimura M**, Ohbuchi K, Sudo Y, Suzuki M, Miyano K, Shiraishi S, Higami Y, Yanagihara K, Hattori T, Kase Y, **Ueta Y**, Uezono Y. Development of ghrelin resistance in a cancer cachexia rat model using human gastric cancer-derived 85As2 cells and the palliative effects of the Kampo medicine rikkunshito on the model. *PLoS One*. 2017 Mar 1;12(3):e0173113. doi: 10.1371/journal.pone.0173113. PMID: 28249026 (2017)
19. So M, Hashimoto H, Saito R, Yamamoto Y, Motojima Y, Ueno H, Sonoda S, **Yoshimura M**, Maruyama T, Kusuhara K, **Ueta Y**. Inhibition of ghrelin-induced feeding in rats by pretreatment with a novel dual orexin receptor antagonist. *J Physiol Sci*. 2017 Jan 4. doi: 10.1007/s12576-016-0517-5. PMID: 28054308 (2017)
20. Matsuura T, Kawasaki M, Hashimoto H, **Yoshimura M**, Motojima Y, Saito R, Ueno H, Maruyama T, Ishikura T, Sabanai K, Mori T, Ohnishi H, Onaka T, Sakai A, **Ueta Y**. Possible Involvement of the Rat Hypothalamo-Neurohypophysial/-Spinal Oxytocinergic Pathways in Acute Nociceptive Responses. *J Neuroendocrinol*. 2016 Jun;28(6). doi: 10.1111/jne.12396. PMID: 27144381 (2016)
21. Matsuura T, Kawasaki M, Hashimoto H, **Yoshimura M**, Motojima Y, Saito R, Ueno H, Maruyama T, Sabanai K, Mori T, Ohnishi H, Sakai A, **Ueta Y**. Effects of central administration of oxytocin-saporin

- cytotoxin on chronic inflammation and feeding/drinking behaviors in adjuvant arthritic rats. *Neurosci Lett*. 2016 May 16;621:104-10. doi: 10.1016/j.neulet.2016.04.010. PMID: 27060190 (2016)
22. Motojima Y, Kawasaki M, Matsuura T, Saito R, **Yoshimura M**, Hashimoto H, Ueno H, Maruyama T, Suzuki H, Ohnishi H, Sakai A, **Ueta Y**. Effects of peripherally administered cholecystokinin-8 and secretin on feeding/drinking and oxytocin-mRFP1 fluorescence in transgenic rats. *Neurosci Res*. 2016 Aug;109:63-9. doi: 10.1016/j.neures.2016.02.005. PMID: 26919961 (2016)
 23. **Yoshimura M**, Ohkubo J, Hashimoto H, Matsuura T, Maruyama T, Onaka T, Suzuki H, **Ueta Y**. Effects of a subconvulsive dose of kainic acid on the gene expressions of the arginine vasopressin, oxytocin and neuronal nitric oxide synthase in the rat hypothalamus. *Neurosci Res*. 2015 Oct;99:62-8. doi: 10.1016/j.neures.2015.05.002. PMID: 26003742 (2015)
 24. Matsuura T, Kawasaki M, Hashimoto H, Ishikura T, **Yoshimura M**, Ohkubo J, Maruyama T, Motojima Y, Sabanai K, Mori T, Ohnishi H, Sakai A, **Ueta Y**. Fluorescent Visualisation of Oxytocin in the Hypothalamo-neurohypophysial/spinal Pathways After Chronic Inflammation in Oxytocin-Monomeric Red Fluorescent Protein 1 Transgenic Rats. *J Neuroendocrinol*. 2015 Jul;27(7):636-46. doi: 10.1111/jne.12290. PMID: 25943916 (2015)
 25. **Yoshimura M**, Matsuura T, Ohkubo J, Maruyama T, Ishikura T, Hashimoto H, Kakuma T, Mori M, **Ueta Y**. A role of nesfatin-1/NucB2 in dehydration-induced anorexia. *Am J Physiol Regul Integr Comp Physiol*. 2014 Jul 15;307(2):R225-36. doi: 10.1152/ajpregu.00488.2013. PMID: 24829503 (2014)
 26. Terawaki K, Sawada Y, Kashiwase Y, Hashimoto H, **Yoshimura M**, Suzuki M, Miyano K, Sudo Y, Shiraishi S, Higami Y, Yanagihara K, Kase Y, **Ueta Y**, Uezono Y. New cancer cachexia rat model generated by implantation of a peritoneal dissemination-derived human stomach cancer cell line. *Am J Physiol Endocrinol Metab*. 2014 Feb 15;306(4):E373-87. doi: 10.1152/ajpendo.00116.2013. PMID: 24347053 (2014)
 27. **Yoshimura M**, Hagimoto M, Matsuura T, Ohkubo J, Ohno M, Maruyama T, Ishikura T, Hashimoto H, Kakuma T, Yoshimatsu H, Terawaki K, Uezono Y, Toyohira Y, Yanagihara N, **Ueta Y**. Effects of food deprivation on the hypothalamic feeding-regulating peptides gene expressions in serotonin depleted rats. *J Physiol Sci*. 2014 Mar;64(2):97-104. doi: 10.1007/s12576-013-0296-1. PMID: 24162946 (2014)
 28. **Yoshimura M**, Matsuura T, Ohkubo J, Ohno M, Maruyama T, Ishikura T, Hashimoto H, Kakuma T, Yoshimatsu H, Terawaki K, Uezono Y, **Ueta Y**. The gene expression of the hypothalamic feeding-regulating peptides in cisplatin-induced anorexic rats. *Peptides*. 2013 Aug;46:13-9. doi: 10.1016/j.peptides.2013.04.019. PMID: 23684922 (2013)

4) Fig 3C: Figure legend mentions a comparison (\$\$) between (one can assume) out-of-phase ZT13 vs ZT13 in-phase? The comparison is missing in the figure. Please, correct it.

The figure has now been updated to include all relevant comparisons, i.e. of effects of treatment (at the same timepoint) and effects of time (within the same treatment type). Please refer to the update figures.

5) Fig 4A and 4C: Units for locomotor activity have to be indicated

Thank you for your attention to detail. We have now updated this, following the standard nomenclature for presentation of activity (Activity (counts/5 min) and body temperature (°C)) in Fig 4a. Locomotor activity shown in Fig 4c is the total

counts in 6h. In the new figure configuration, we have re-organized this figure (now Figure 6) for better readability, which are now displayed in Fig. 6a and 6c.

6) Statistics: Tests used to probe normality before ANOVA analysis should be mentioned and information about the test results have to be included at least in the figure legend to assess the size of the differences observed.

Thank you for your indication. We have now tested all datasets for normality, before re-running the ANOVA analyses. Tests used to probe for normal distribution were D'Agnostino & Pearson, Anderson-Darling, Shapiro-Wilk and Kolmogorov-Smirnov tests. All datasets were found to pass normality test ($\alpha=0.05$) in at least one test. This information has been added to the methods section of the manuscript.

Where two or more treatment conditions (e.g. *in-phase* versus *out-of-phase* infusion) were being assessed over two or more timepoints, then 2-WAY ANOVA was used to assess main effect of treatment, main effect of time, and interaction between time and treatment. Details of the results for these statistical analyses have now been included in the figure legends. Please also see in lines 609-625, on pages 26 and 27.

7) Figure legends contain a description of the results, which might be redundant with the result section. As mentioned above, would be more informative to have in the legend the results of the statistical tests.

The figure legends have been updated according to the reviewer's suggestion.

8) Page 10 line 228: Authors referred to a change in appetitive behavior during the inactive phase, however only food intake was measured every 12 hs and the correlation with the change is CORT peak might be not happening at the same time. Please, revise this statement.

The reviewer makes an interesting point, which we thought was worth pursuing further so we did an additional experiment to test whether we could detect food intake differences at a better resolution than 12 hourly. The graph below shows food intake every 6 hrs. While there appears to be a trend that may support a slightly bigger difference in food intake (between *in-phase* and *out-of-phase* treatment groups) at ZT0-ZT6 than ZT6-ZT12, as well as a trend that may support a slightly bigger difference in food intake (between *in-phase* and *out-of-phase* treatment groups) at ZT12-ZT18 than ZT18-ZT24, there was no statistical

difference between amount of food ingested during ZT0-ZT6 and ZT6-ZT12 (within treatment) nor between ZT12-ZT18 and ZT18-ZT24 (within treatment). Therefore, while we include the data here for the reviewers' benefit, we don't think the paper would benefit from its addition. We hope you agree that our new data from the additional experiment explains this phenomenon more elegantly, than showing this data in the paper.

We have also revised the statement in the discussion according to the reviewer's indication. Please see in lines 313-317, on page 14.

9) Authors suggested a role of GR however, is it possible to exclude MR-dependent effects?

It is not possible at this stage to exclude MR-dependent effects. This possibility has now been discussed. Please see in lines 335-381, on pages 15-17.

10) Page 14 line 328-330: this sentence needs references

References are now added.

11) Page 14 line 332-333: Authors need to explain what they mean with "rational approach improved therapeutics"

Thank you for pointing this out. We agree that it this phrase is poorly worded, it has now been revised to 'Our data contribute to improved understanding of the mechanisms underlying these metabolic effects and should help provide a framework for a rational approach for developing improved therapeutics.'

12) More elaborated discussion on the changes observed in core body temperature with no changes in activity is necessary.

An additional experiment has been performed, and the data added to Supplementary Figures. Please see lines 429-439, on pages 19.

13) Levels of peripheral hormones regulating food intake might be necessary to really explain whether the effects on food intake are directly or indirectly caused by CORT (page 12 lines 276-278). If possible, collecting these data should be considered.

We agree with your suggestion. Actually, feeding behavior is known to be controlled by many factors, including peripheral hormone, brain neuropeptides, and/or light-dark cues. Investigating peripheral feeding regulating hormones, such as GLP-1 or ghrelin is of interest for hypothesizing the underlying mechanism. It would have been interesting to measure peripheral hormones, as you suggested, but our financial situation did not allow for these types of additional experiments. Therefore we had to make some strong decisions about which experiments to prioritize for our resubmission. As our research question focused more on the central regulation of feeding behavior by hypothalamic regulatory neuropeptides, which we hypothesized to be directly regulated by NPY. We prioritized the additional experiment to test whether CORT-activation of GR directly modulates the transcription of the NPY gene in the arcuate nucleus (Fig 8).

14) In order to attribute to CORT the changes in feeding rhythms and metabolic phenotype authors would need to investigate food intake at a later time point after surgery. Could be possible that SCN driven rhythms become more dominant later on and restore the feeding rhythms?. If possible, collecting these data should be considered.

Yes the reviewer is quite correct. Over time, in a natural environment, circadian rhythms will entrain to a new lighting schedule. This is most obvious with transmeridian travel. At first, the misalignment is very pronounced in the phenomenon we know as 'JetLag'. The internal clocks can then re-entrain to the new schedule, shifting by 1 hour per day. For example, it takes 10 days to re-entrain to a 10 hour time shift. This is already very well appreciated. Interestingly, it is widely thought that ensuring meal times coincide with the new schedule helps entrain circadian clocks more efficiently. Although there is no evidence for entrainment faster than 1hour/day. What is unique in our study is that we are experimentally 'clamping' the CORT pattern out-of-phase with the lighting for 5 days. There was no evidence of feeding pattern renormalizing over this time.

Whether SCN rhythms become more dominant later on and restore feeding rhythms is an interesting question, but one we cannot answer with our current methodology. We are restricted by UK Home Office ASPA regulations that have permitted only sub-chronic treatments on our Procedural Project Licence.

15) Please, add proper reference to supplementary figures and supplementary figures legends.

All figures, supplementary figures have now been properly referenced.

16) Control group has to be called either control or sham in a consistent way throughout the paper

This has now been updated throughout the manuscript for consistency. We have explained what the control group is (ie sham-operated (adrenal-intact) rats) when first described in lines 99-100 on page 5, then used the term control consistently thereafter.

17) Page 4 line 76: POMC and CART abbreviations need to be explained

This has been updated.

18) Page 4 line 79 and 82: NPY abbreviation was already explained above

This has been updated.

19) Page 4 line 86: should read "current study"

This has been updated.

20) Page 4 line 87: should read "we investigate.....associated with phase shifting circadian glucocorticoid profile" since there is no shift work or transmeridian travel effects assessed.

We have revised this sentence accordingly to "In our current study we investigate the mechanisms underlying the adverse metabolic effects associated with phase shifting the circadian glucocorticoid profile". Please see in lines 84-86 on page 4.

21) Page 5 line 114: "highly disordered" food intake sounds inappropriate to describe a 12-h time resolution in food intake.

As you suggested, we have changed the description into “Interestingly, rats in the *out-of-phase* group ate continuously, even during the circadian inactive period.” Please see in lines 316-317 on page 14.

22) Page 7 lines 164 and 165: Consistent is written twice in two consecutive sentences.

We agree with the reviewer that we have overused this phrase, therefore we have now removed/replaced the word ‘consistent’, in lines 187, 320, and 343, leaving it in lines 35, 346, and 406.

23) Color coding could be confusing. As a suggestion, out-of-phase group could be either red or black and red could be avoided for the IHH results might help.

As you suggested, it is better to avoid using red column since red colour was used in “out-of-phase” in Fig 1b and 1d. I changed the color of the column of the figure. Please see Fig. 5b.

24) Page 17 line 391: cort should read CORT

We have revised “cort” into “CORT”.

Reviewer #2 (Remarks to the Author):

This manuscript by Mitsuhiro Yoshimura et al. examined the effects of Phase-shifting the circadian glucocorticoid profile on energy metabolism and feeding behavior in rats to reveal the relationship between circadian glucocorticoid and hypothalamic neuropeptide gene in the process of eating disorder. Besides, the authors have shown that appropriate timing of food intake may be associated with the observed normal circadian profile of CORT secretion.

Overall this is a straightforward study on a fascinating topic. Additionally, there are several advantages of circadian on energy metabolism shown in the last several years, and this one is a continuation of one of such studies. I have the following reservations with this study, and they had to be addressed.

1. It seems like these authors have selectively picked up few genes without providing too much logistics and background. The logistics checking

anorexigenic and orexigenic neuropeptide gene expression level is not defined appropriately.

The reason we chose these feeding regulating neuropeptides' genes is because they have been shown previously to regulate appetitive behaviour and energy balance. The rationale for our interest in these genes are described in the introduction (Lines 71-83). In addition, the antisense probe sequences of these genes have been confirmed and validated by our previous studies, and others (see the provided extensive reference list for ISH performed by Yoshimura et al), so we could feel confident in the findings, which is always important. As you suggested, there are many feeding regulating genes in addition to the genes we investigated. Although we agree with your suggestion, we stand by our choice of targets focusing on the specific feeding regulating gene expression in the present study.

2. The authors have also jumped the here and there without providing any mechanistic detail. For example, they are increasing NPY, AgRP, POMC, and CART expression in brain. What is the relationship of CORT with these genes?

This has now been addressed in detail, with data from the new experiment and added discussion. Please see in lines 38-40 on page 2, lines 101-116 on page 5, and lines 258-280 on pages 11-12. Methods were also added in lines 543-607 on pages 24-26.

3. Additionally, it will be essential to demonstrate cytological evidence to reveal the impact of CORT disorder, for example, mitochondrial function/integrity, oxidative stress. This might help to demonstrate the mechanistic effect of disordered feeding behaviour for the obesity models.

Indeed, we agree with the reviewer that it will be interesting to assess oxidative stress and mitochondrial function/integrity to more fully understand the impact of CORT disorder, especially in the pathological model of obesity. However, we don't necessarily agree that this will demonstrate the mechanism of disordered feeding behaviour for the obesity models. Oxidative stress is affected directly by glucocorticoids, again as a GR-mediated transcriptional regulation of genes including Cytochrome C Oxidase and NADH:ubiquinone oxidoreductase. While we agree that this type of mitochondrial and oxidative stress disorder is likely a major part of CORT disorders, the mechanism that we believe is responsible for

the disordered feeding behaviour is via direct actions of CORT via GR on NPY regulation in the ARC, as well as related but indirect actions via energy balance, as explained in our discussion. Please see in lines 335-381 on pages 15-17.

4. Several places author has indicated that Phase-shifting the circadian glucocorticoid profile induces food intake, yet the mechanisms underlying its detailed effects are not fully understood. I have failed to see any mechanism in this paper. The authors in tone in the entire manuscript about these kinds of statements.

We have now performed an additional experiment to address this point. Using chromatin immunoprecipitation assays, we demonstrate that the orexigenic neuropeptide NPY is regulated at the transcriptional level by an acute treatment with CORT, even during the circadian nadir (ZT2). We found that GR is recruited to a GRE site upstream of the NPY gene in a CORT-dependent manner in ADX rats treated with a s.c. injection of either CORT-HBC-saline or HBC-saline (vehicle control), and that pSer5 RNA Pol2 occupancy of the NPY gene TSS is concomitantly increased in a CORT-dependent manner. Further support for CORT-responsiveness of the arcuate nucleus (ARC), is evident in the CORT-induced GR binding at the well-characterised hypersensitive Per1 GRE, with concomitant increase in pSer5 RNA Pol2 occupancy of the Per1 gene TSS. Taken together, these data are consistent with our premise that exposure to elevated GC levels, even during the circadian nadir can over-ride circadian influence in the ARC, in a similar manner to demonstrated in other GR-responsive parts of the brain (Birnie et al, *Proc Natl Acad Sci USA*, 2023). Please see our relevant results and discussion in lines 256-280 on pages 11-12, and in lines 335-381 on pages 15-17, respectively. Please also see new Fig. 8.

REVIEWERS' COMMENTS:

Reviewer #1 (Remarks to the Author):

The article entitled "Phase-shifting the circadian glucocorticoid profile induces disordered feeding behavior by dysregulating hypothalamic neuropeptide gene expression" by Yoshimura and colleagues describes feeding behavior in adrenalectomized rats infused with corticosterone in- and out-of-phase with circadian lighting cues.

The authors have made a considerable effort to improve the manuscript according to this reviewer suggestions. However, further changes are suggested before considering it for publication.

Major concerns:

1) In the abstract, line 41: the circadian system is one (not two). Authors should consider changing the wording here to "misaligned cues" or "anti-phasic cues" or "out of synchrony cues"

2) In the introduction, line 71: The sentence "For example, exogenous chronic CORT treatment in mice, delivered ad libitum in drinking water, resulted in high CORT levels at both the circadian active and inactive phases" misses two things: a reference and the fact that when CORT is delivered in the drinking water, the circulating levels will most likely mirror the drinking behaviour, which is higher during the active phase and lower during the inactive phase. Please, rephrase and add a reference.

3) In the introduction, lines 71-75: The authors should at least mention the anorexigenic/orexigenic effects of the neuropeptides mentioned to make a stronger point on the meaning of the previous results.

4) In the last introduction sentence (line 116), the results section (line 118), the discussion section (line 283) and probably somewhere else: LD cycle cannot be called circadian cue; it is strictly an environmental daily cue. Please, clarify this concept throughout the text.

6) The experiment to show GR binding to GREs in Per1 and NPY promoters is informative and it shows that both genes could be modulated by CORT but it does not demonstrate necessity or sufficiency. Authors should avoid overinterpretation of the data.

7) In the discussion section (line 315-317): the statement concluding that the out of phase group ate continuously and that this might indicate that food intake is driven by both "lights off" and CORT is misleading. It could just only mean that when the daily light cues are misaligned to CORT levels, the food intake rhythms are dampened. Please, consider a more thorough interpretation of these data.

8) In the discussion section (line 317-376): this reviewer's opinion is that this paragraph should be shortened reducing personal views on other scientist's work and providing only that information that helps the discussion of the data.

9) In the discussion section (line 403-405) and likely somewhere else: it is an over interpretation of the data to say that circadian rhythms of neuropeptides remain intact when only two time points were measured in entrained conditions. Please, carefully checked the whole manuscript for misleading concepts like this one.

10) Regarding the answer to comment 3 from the previous revision round. This reviewer considers that a negative control has to be run always for each single experiment without relying on previous experiments/publications.

Reviewer #2 (Remarks to the Author):

The authors correctly addressed the review and corrected the manuscript. In addition, they also clarified all doubts. In my opinion, the manuscript can be published in its current form.

We thank the reviewers for their detailed and insightful comments on our revised manuscript. We have made all changes recommended. *Details in text below.*

1) In the abstract, line 41: the circadian system is one (not two). Authors should consider changing the wording here to “misaligned cues” or “anti-phasic cues” or “out of synchrony cues”

Line 41 has been amended as per the reviewer’s suggestion: Taken together, our data highlight the adverse behavioural outcomes that can arise when two circadian systems have anti-phasic cues, in this case impacting on the glucocorticoid-regulation of a process as fundamental to health as feeding behaviour.

2) In the introduction, line 71: The sentence “For example, exogenous chronic CORT treatment in mice, delivered ad libitum in drinking water, resulted in high CORT levels at both the circadian active and inactive phases” misses two things: a reference and the fact that when CORT is delivered in the drinking water, the circulating levels will most likely mirror the drinking behaviour, which is higher during the active phase and lower during the inactive phase. Please, rephrase and add a reference.

Lines 73-74 have been amended as per the reviewer’s suggestion: ‘For example, exogenous chronic CORT treatment in mice, delivered *ad libitum* in drinking water, resulted in high CORT levels during the daily active phase¹², reflecting increased drinking behaviour during the active phase, as previously shown (new reference Stamper et al 2015 Stress 18(1):76-87).

3) In the introduction, lines 71-75: The authors should at least mention the anorexigenic/orexigenic effects of the neuropeptides mentioned to make a stronger point on the meaning of the previous results.

We appreciate the feedback and the thoughtful insight you have shared. We agree that elaborating on the anorexigenic and orexigenic effects of the neuropeptides mentioned would indeed enhance the significance of the observed results. By discussing the potential roles of these neuropeptides in regulating appetite and energy balance, we could provide a more comprehensive context for interpreting the findings related to hyperphagia and obesity.

Lines 79-80: “AgRP and NPY are known for their role in promoting feeding behavior, whereas POMC and CART are associated with satiety.”

4) In the last introduction sentence (line 116), the results section (line 118), the discussion section (line 283) and probably somewhere else: LD cycle cannot be called circadian cue; it is strictly an environmental daily cue. Please, clarify this concept throughout the text.

The term ‘circadian’ has been amended to either ‘environmental daily cues’, ‘time of day’, ‘daily light cues’, ‘daily’ or ‘AM/PM differences’ throughout the text.

In some other cases, the word ‘circadian’ was superfluous, for example ‘the circadian active phase’ or ‘circadian inactive phase’. Therefore the word was simply deleted in these case.

NB The term ‘circadian’ has been retained when referring to phenomenon or data that has been taken with frequent measurements (5 or 10 minutely). For example when referring to ‘Circadian glucocorticoid profile’ and ‘Circadian core body temperature (CBT).

6) The experiment to show GR binding to GREs in *Per1* and NPY promoters is informative and it shows that both genes could be modulated by CORT but it does not demonstrate necessity or sufficiency. Authors should avoid overinterpretation of the data.

Lines 119-123 amended to 'Taken together, these data support a direct glucocorticoid-dependent transcriptional regulation of NPY similarly to that previously shown for *Per1*. This supports our hypothesis that expression of NPY can become dysregulated during *out-of-phase* CORT exposure resulting in orexigenic behaviour independent of environmental daily cues.'

7) In the discussion section (line 315-317): the statement concluding that the out of phase group ate continuously and that this might indicate that food intake is driven by both "lights off" and CORT is misleading. It could just only mean that when the daily light cues are misaligned to CORT levels, the food intake rhythms are dampened. Please, consider a more thorough interpretation of these data.

Lines 328-330: Amended to 'Although CORT exposure was expected to induce feeding, food intake still occurred during the active phase, potentially indicating that when daily light cues are misaligned to CORT levels there is a continued drive to eat throughout the inactive as well as the active phase.'

8) In the discussion section (line 317-376): this reviewer's opinion is that this paragraph should be shortened reducing personal views on other scientist's work and providing only that information that helps the discussion of the data.

Lines 361-368 addresses GR regulation of AGRP, and the limitation in our ability to assess this in a meaningful way in our experimental design. We therefore think this discussion should be retained but can remove it if the editor thinks the discussion is too lengthy.

Lines 375-388 addresses the potential role of MR, which was a point raised by reviewers. We think this discussion should be retained.

Lines 369-374 has now been revised in light of the reviewer's suggestion: 'Our data clearly demonstrates corticosterone-inducible GR binding at a GRE site in the NPY promoter and concomitant increase in enrichment of pSer5 RNA Pol2 at the NPY TSS. Although there is data from the GRdim mouse suggesting NPY regulation may not need GR²⁷, more recent data has revealed that the GRdim model is in fact functional in certain contexts²⁸⁻³¹, which is further supported by our current data.'

9) In the discussion section (line 403-405) and likely somewhere else: it is an over interpretation of the data to say that circadian rhythms of neuropeptides remain intact when only two time points were measured in entrained conditions. Please, carefully checked the whole manuscript for misleading concepts like this one.

Thank you for this suggestion, wording has been changed to 'AM/PM differences' when only two timepoints were assessed.

10) Regarding the answer to comment 3 from the previous revision round. This reviewer considers that a negative control has to be run always for each single experiment without relying on previous experiments/publications.

Yes, we agree. We will certainly do so in future!

Additionally, in lines 457-458 and lines 495-504, we have updated supplier

information and details about the CORT and ACTH assay methods.

Reviewer #2 (Remarks to the Author):

The authors correctly addressed the review and corrected the manuscript. In addition, they also clarified all doubts. In my opinion, the manuscript can be published in its current form.

Thank you!

Additionally, the following changes have been made according to the CommsBio_AIP_table.

- Excel spreadsheet of all raw data uploaded as Supplementary Data.
- Abstract amended to present tense, and final sentence including broader impacts and how this research will be used in the future. *'Here we demonstrate, in rodents, how the timing of feeding behaviour becomes disordered when circulating glucocorticoid rhythms are dissociated from lighting cues; a phenomenon most commonly associated with shift-work and transmeridian travel 'jetlag'. Adrenalectomized rats are infused with physiological patterns of corticosterone modelled on the endogenous adrenal secretory profile, either in-phase or out-of-phase with lighting cues. For the in-phase group, food intake is significantly greater during the rats' active period compared to their inactive period; a feeding pattern similar to adrenal-intact control rats. In contrast, the feeding pattern of the out-of-phase group is significantly dysregulated. Consistent with a direct hypothalamic modulation of feeding behaviour, this altered timing is accompanied by dysregulated timing of anorexigenic and orexigenic neuropeptide gene expression. For Neuropeptide Y (Npy), we report a glucocorticoid-dependent direct transcriptional regulation mechanism mediated by the glucocorticoid receptor (GR). Taken together, our data highlight the adverse behavioural outcomes that can arise when two circadian systems have anti-phasic cues, in this case impacting on the glucocorticoid-regulation of a process as fundamental to health as feeding behaviour. Our findings further highlight the need for development of rational approaches in the prevention of metabolic dysfunction in circadian-disrupting activities such as transmeridian travel and shift-work.'*
- All gene and mRNA names have been updated to official gene symbol, and are in italics.
- Fewer abbreviations have been used.
- The two tables have been removed; with a Figure (in the case of Table 1) and probe sequences are now listed in the methods (in place of Table 2).
- The Methods now include a separate section titled "Statistics and Reproducibility"
- The following statement "We have complied with all relevant ethical regulations for animal testing." has been included.
- 'Competing interests' statement has been amended.